# Holographic Node Representations: Pre-training Task-Agnostic Node Embeddings

**Beatrice Bevilacqua**
Purdue University
bbevilac@purdue.edu

**Joshua Robinson**
Stanford University
joshrob@cs.stanford.edu

**Jure Leskovec**
Stanford University
jure@stanford.edu

**Bruno Ribeiro**
Purdue University
ribeirob@purdue.edu

## Abstract

Large general purpose pre-trained models have revolutionized computer vision and natural language understanding. However, the development of general purpose pre-trained Graph Neural Networks (GNNs) lags behind other domains due to the lack of suitable generalist node representations. Existing GNN architectures are often tailored to specific task orders, such as node-level, link-level, or higher-order tasks, because different tasks require distinct permutation symmetries, which are difficult to reconcile within a single model. In this paper, we propose *holographic node representations*, a new blueprint for node representations capable of solving tasks of any order. Holographic node representations have two key components: (1) a task-agnostic expansion map, which produces highly expressive, high-dimensional embeddings, free from node-permutation symmetries, to be fed into (2) a reduction map that carefully reintroduces the relevant permutation symmetries to produce low-dimensional, task-specific embeddings. We show that well-constructed expansion maps enable simple and efficient reduction maps, which can be adapted for any task order. Empirical results show that holographic node representations can be effectively pre-trained and reused across tasks of varying orders, yielding up to 100% relative performance improvement, including in cases where prior methods fail entirely.

## 1 Introduction

The development of node embeddings capable of effectively solving multiple tasks remains an unsolved problem in graph learning (Mao et al., 2024). Just as LLMs have demonstrated the ability to generate versatile representations that can be adapted for a variety of downstream tasks (Radford et al., 2019; Gemini Team et al., 2023; OpenAI, 2023), there is a growing need for graph models that can produce embeddings versatile enough to solve new tasks not foreseen at train time. Despite significant advancements in Graph Neural Networks (GNNs), most existing approaches remain highly specialized, which limits their broader applicability across diverse graph tasks.

Existing embeddings are typically designed for specific types of tasks, or "task orders", and often struggle with others. For example, embeddings optimized for node classification (an order-1 task) may not work well for link prediction (an order-2 task) (You et al., 2019; Srinivasan & Ribeiro, 2020; Zhang et al., 2021), and vice-versa. This challenge extends to higher-order tasks, and embeddings intended to handle various tasks generally fail to perform well on any task. A crucial yet often overlooked reason for this failure is that different task orders possess different permutation symmetries, which are difficult to program into a single model. *This order-specificity of existing models hinders the development of general pre-trained node representations that can be adapted to any graph task.*

In this work, we revisit the notion of node embeddings to create *holographic node representations*, a new type of node representations specifically designed to be pre-trained and then adapted to any task order. The key innovation is a new treatment of node permutation symmetries. Instead of immediately introducing potentially undesirable symmetries, as is commonly done with standard permuta-

tion equivariant GNNs (Bronstein et al., 2021), holographic node representations use the majority of their computation to learn an *expansion map* which produces highly-descriptive symmetry-free representations. These are then fed into a *reduction map*, which introduces task-specific symmetries to produce embeddings tailored for the task. The reduction map is lightweight, allowing it to be efficiently adapted and learned for each new task order. The core idea is that, regardless of the pre-training task order, a suitable reduction map can always be constructed and learned anew for new tasks of any order, leveraging the output of the expansion map as the pre-trained node embeddings.

In addition to introducing a suitable notion of representations for order-agnostic embeddings, we present the first practical instantiation of this concept, which we call HoloGNN. The key technical challenge we address is *how* to add and remove permutation symmetries. For the symmetry-free expansion map, we combine a standard message passing model with repeated symmetry breaking by perturbing node features, producing a *sequence* of flat embeddings for each node, each able to capture different information fragments. We demonstrate that this design is provably able to express eigenvectors without sign or basis ambiguities, which ensures HoloGNN is able to capture rich graph features (Belkin & Niyogi, 2003). The reduction map fuses together the sequence of representations for task-specific groups of nodes to output any-order (node-, link-, and higher-order) representations. We also prove that our combination of expansion and reduction maps produces maximally-expressive representations from a permutation symmetry perspective.

We experimentally validate our approach, showing that HoloGNN is able to learn representations that transfer between task orders consistently and significantly outperforming the baselines. For instance, on the CORA dataset, the performance of HoloGNN decreases by only $1.5\%$ when pre-trained on link prediction and adapted to node classification, while standard GNNs show a $7\%$ performance drop, and link-prediction models such as NBFNet (Zhu et al., 2021) and SEAL (Zhang & Chen, 2018), experience a significantly larger drop of up to $41\%$.

In summary, our key contributions are:

(1) Introducing *holographic node representations*, a novel approach that allows node embeddings to be pre-trained on tasks of any order and then be efficiently adapted to solve new tasks of new orders by respecting the task-specific permutation symmetries;

(2) Presenting HoloGNN, the first practical architecture for learning holographic node representations, addressing the technical challenge of adding and removing permutation symmetries;

(3) Theoretically and empirically demonstrating that holographic node representations offer significant improvements in generalization to new tasks, providing highly expressive representations that consistently outperform existing methods, irrespective of the pre-training task order.

**Notation.** Let $G = (\mathbf{A}, \mathbf{X})$ be a graph, where $\mathbf{A} \in \{0, 1\}^{n \times n}$ is the adjacency matrix and $\mathbf{X} \in \mathbb{R}^{n \times d_x}$ the node features. We write $v \in [n] = \{1, \ldots, n\}$ to denote a node in $G$, or equivalently $v \in V$. For node embeddings $f(\mathbf{A}, \mathbf{X}) \in \mathbb{R}^{n \times d}$, $f(\mathbf{A}, \mathbf{X})_v$ denotes the embedding in $\mathbb{R}^d$ of node $v \in [n]$. Similarly, for $f(\mathbf{A}, \mathbf{X}) \in \mathbb{R}^{n \times T \times d}$, $f(\mathbf{A}, \mathbf{X})_v$ denotes the $\mathbb{R}^{T \times d}$ slice. We denote the permutation group on $[n]$ by $\mathbb{S}_n$, which is the collection of all bijective maps $[n] \to [n]$. In general, for tensors $\mathbf{X} \in \mathbb{R}^{\binom{n}{r} \times d_x}$, we write $\mathbf{X}_S \in \mathbb{R}^{d_x}$ for a set $S \subseteq [n]$ of cardinality $r$. For $S$, we define $\pi \circ S = \{\pi(i) \mid i \in S\}$, and write $\pi \circ \mathbf{X}$ to denote the permutation of the rows of $\mathbf{X}$ by applying $\pi$ to set $S$ indexing them, with $\circ$ the permutation action. Finally, $2^{[n]}$ denotes the power set of $[n]$.

## 2    PERMUTATION SYMMETRIES AND TASK GENERALIZATION

An often overlooked aspect of graph learning is the task-specific nature of existing embeddings, typically designed for a particular task "order", which is the size of the set of nodes involved in the predictions. For example, node classification involves predictions for individual nodes (order-1), while link prediction for pairs of nodes (order-2). Since there are many important graph tasks of different orders, there is a need for powerful models producing embeddings suitable for all orders.

However current embeddings are task-order specific. One reason for this is that many existing embeddings are *structural*, meaning that the embeddings do not depend on the IDs of each node. Crucially, the exact permutation group describing this invariance is not universal, but is inherently dependent on the task order. This is clearly reflected in the fact that the cardinality of the required

embeddings for structural representations is determined by the task order itself, as can be seen in the definition of structural representations.

**Definition 2.1** (Structural Representations). Let $S$ be a set of $r$ nodes of a graph $G = (\boldsymbol{A}, \boldsymbol{X})$. A function $f : \{0, 1\}^{n \times n} \times \mathbb{R}^{n \times d_x} \to \mathbb{R}^{\binom{n}{r} \times d}$ returns structural representations of order $r$ if $f$ is equivariant to permutations of the nodes, that is if $f(\boldsymbol{A}, \boldsymbol{X})_S = f(\pi \circ \boldsymbol{A}, \pi \circ \boldsymbol{X})_{\pi \circ S}$, for any $\pi \in \mathbb{S}_n$.

In other words, a function returns structural representations if it produces embeddings that are equivariant under any permutation of the node ordering. Within this class, there is a clean notion of most-expressive representation, which occurs when isomorphic node sets share the same representation (necessitated by Definition 2.1), and all non-isomorphic sets have distinct representations.

**Definition 2.2** (Most-Expressive Structural Representations). A function $f : \{0, 1\}^{n \times n} \times \mathbb{R}^{n \times d_x} \to \mathbb{R}^{\binom{n}{r} \times d}$ returns most-expressive structural representations of order $r$ if for every graphs $G = (\boldsymbol{A}, \boldsymbol{X})$, $G' = (\boldsymbol{A}', \boldsymbol{X}')$ and sets of nodes $S$, $S'$ of cardinality $r$, $f(\boldsymbol{A}, \boldsymbol{X})_S = f(\boldsymbol{A}', \boldsymbol{X}')_{S'}$ iff $\exists \pi \in \mathbb{S}_n$ such that $\boldsymbol{A} = \pi \circ \boldsymbol{A}'$, $\boldsymbol{X} = \pi \circ \boldsymbol{X}'$, $S = \pi \circ S'$.

A most-expressive structural representation is sufficient to solve any task of that order (Srinivasan & Ribeiro, 2020). However, in general, the most expressive structural representation of one order is not also the most expressive structural representation of another order. For instance, Srinivasan & Ribeiro (2020) and Zhang et al. (2021) prove that a most-expressive structural representation of order-1 tasks cannot serve as a most-expressive structural representation for order-2 tasks, as show in Figure 1. Because of this, it is not possible to pre-train a single structural representation that is most-expressive and therefore able to solve tasks of any order.

Figure 1: Nodes $v_2$ and $v_3$ are isomorphic; links $(v_1, v_2)$ and $(v_4, v_3)$ are isomorphic; links $(v_1, v_2)$ and $(v_1, v_3)$ are **not** isomorphic. However, if we aggregate most-expressive node representations as the link representation, $(v_1, v_2)$ and $(v_1, v_3)$ will get the same prediction. Example from Zhang et al. (2021).

In addition to structural representations, positional ones, such as those derived from eigenvectors (Belkin & Niyogi, 2003; Dwivedi et al., 2023), play an important role in graph learning. Unlike structural embeddings, positional embeddings assign distinct embeddings to isomorphic node sets. While positional embeddings can be highly effective for tasks requiring the unique placement of nodes within a graph, they tend to underperform on others (Srinivasan & Ribeiro, 2020). For instance, positional embeddings often perform well in link prediction tasks, where the relative positions of the node endpoints are critical (Lim et al., 2023). However, they are less effective for node tasks, where node similarity should reflect structural similarity rather than positional proximity. In other words, current positional representations are also task specific.

**Challenges for learning a single embedding for different tasks.** Because structural and positional representations are task specific, they are unsuitable output candidates for pre-training a single model that produces representations capable of solving tasks of different orders. This claim can be formulated precisely for node embeddings, which are perhaps the most important class of embeddings due to their linear scalability in the graph size, which is essential when scaling to large graphs.

**Proposition 2.3** (Informal). *For any node embedding model $f$, there exists two tasks of different orders for which at least one is not solvable using $f$.*

The formal version of Proposition 2.3 is presented in Appendix B.1 as Proposition B.1. This impossibility result highlights why graph machine learning has continued to rely on specialized architectures tailored for specific tasks, unlike the more generalized pre-trained models seen in NLP and computer vision. It also clarifies that there is a need for novel node representations that extend beyond conventional node embeddings. In the next section, we present our approach bypassing this impossibility result by representing nodes with sequences of (flat) node embeddings.

## 3 HOLOGRAPHIC NODE REPRESENTATIONS

The previous section demonstrated the need for a new notion of node representations beyond the typical single "flat" embedding vector per node. Our proposal is to use *holographic node representations*. Holographic representations have existed outside of the graph learning context since the

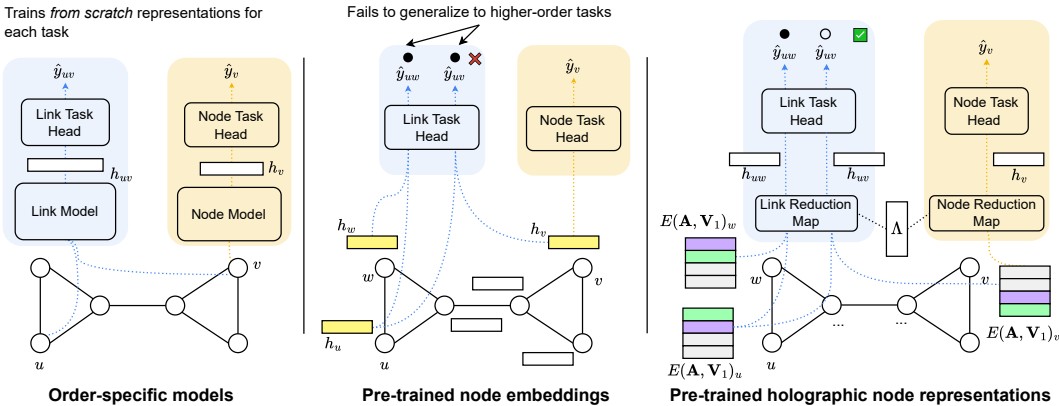

Figure 2: While order-specific models must be retrained for each new task, holographic node representations can be adapted to new tasks. Pre-trained (structural) node embeddings cannot solve high-order tasks since isomorphic nodes get the same representation, making it impossible to differentiate non-isomorphic links. Instead, holographic node representations generate 2D node representations ($T \times d_e$), where the additional $T$ dimension captures multiple symmetry-free graph views (so isomorphic nodes can be distinguished in at least one view). The reduction map reduces these representations back to 1D, using $\Lambda$ to re-introduce the correct permutation symmetries for the task.

work of Feldman (1986) and Hinton (1990), which studied how to store complex concepts in neural networks. At one extreme was the one-concept, one-neuron idea, and at the other was holographic representations which distributed individual concepts across neurons. These holographic representations worked by producing representations conditional on different *views* of the data, rather than relying on a single "absolute" representation. In this section, we adapt this principle to graph data, proposing *holographic node representations*, which produce representations conditioned on different nodes. These conditional representations overcome the impossibility result in Proposition 2.3.

Holographic node representations are defined through an *expansion* map and a *reduction* map (Figure 2). The expansion map produces task-agnostic high-dimensional representations of the individual units of interest (in our case each node) that capture different permutation-sensitive projections of the object. The permutation-sensitivity of these projections is designed in a way that we can always create a reduction map (a light-weight task-specific map) that reintroduces the relevant symmetries for the task at hand, producing structural representations that match the task order. In this way, regardless of the pre-training task order, a suitable reduction map can always be learned for any new task of any order, using the output of the expansion map as the pre-trained node embeddings.

Holographic node representations start with structural node representations,

$$\boldsymbol{V}_1 = g_{\text{struc}}(\boldsymbol{A}, \boldsymbol{X}) \in \mathbb{R}^{n \times d_1}, \tag{1}$$

where $g_{\text{struc}}$ is a learnable function returning structural node representations. This function can be as simple as a single-layer GNN or even the identity function ($\boldsymbol{V}_1 = \boldsymbol{X}$). Importantly, the expressiveness of our approach is not limited by that of $g_{\text{struc}}$, since lost distinctions between nodes can be reintroduced in the expansion map. But formulating $\boldsymbol{V}_1$ as a general structural representation has the benefit of allowing the forthcoming expansion map to use prior efforts to obtain rich representations.

**Definition 3.1** (Holographic Node Representations). *Holographic node representations* consist of two learnable, parameterized maps:

(1) **Expansion Map:**

$$E_\theta : \{0,1\}^{n \times n} \times \mathbb{R}^{n \times d_1} \to \mathbb{R}^{n \times T \times d_e} \times (2^{[n]})^L \times \mathbb{N}^T \tag{2}$$

The expansion map, parameterized by $\theta$, takes as input the adjacency matrix and the initial structural representations, and it outputs: (a) A ($T \times d_e$)-dimensional representation for each of the $n$ nodes (2D representation), denoted by $V_\theta(\boldsymbol{A}, \boldsymbol{V}_1)$; (b) A sequence of $L$ lists of node IDs, where nodes within each list share the same role, and nodes in different lists have distinct roles; (c) A sequence of integers, indicating how the $T$ node representations should be grouped.

(2) **Reduction Map:**

$$R_\psi : \mathbb{R}^{n \times T \times d_e} \times (2^{[n]})^L \times \mathbb{N}^T \to \mathbb{R}^{\binom{n}{r} \times d_r} \qquad (3)$$

The reduction map, parameterized by $\psi$, takes the output of the expansion map and produces 1D representations for any set of $r$ nodes.

The mappings satisfy the following properties:

Property (1): The composition $R_\psi \circ E_\theta$ produces $\binom{n}{r}$ structural representations (one for each set of $r$ nodes), i.e., $\pi \circ R_\psi(E_\theta(\boldsymbol{A}, \boldsymbol{V}_1)) = R_\psi(E_\theta(\pi \circ \boldsymbol{A}, \pi \circ \boldsymbol{V}_1))$ for any $\pi \in \mathbb{S}_n$.

Property (2): For any undirected graph $G = (\boldsymbol{A}, \boldsymbol{X})$ and isomorphic nodes $u, v \in G$, with $u \neq v$ and having different neighborhoods, there exists a $\theta$ such that, $V_\theta(\boldsymbol{A}, \boldsymbol{V}_1)_v \neq V_\theta(\boldsymbol{A}, \boldsymbol{V}_1)_u$. That is, the expansion map $E_\theta$ can distinguish isomorphic nodes.

Note that both $E_\theta$ and $R_\psi$ are neural networks, with parameters $\theta, \psi$, for which we introduce explicit designs in Section 4. From now on, we will write $E$ and $R$ instead of $E_\theta$ and $R_\psi$ for simplicity when it is unambiguous. Additionally, we denote the three outputs of the expansion map as $E_\theta(\boldsymbol{A}, \boldsymbol{V}_1) = \left(V(\boldsymbol{A}, \boldsymbol{V}_1), (h_\lambda(\boldsymbol{A}, \boldsymbol{V}_1))_{\lambda=1}^L, \Lambda\right) \in \mathbb{R}^{n \times T \times d_e} \times (2^{[n]})^L \times \mathbb{N}^T$.

## 3.1 INTERPRETING THE DEFINITION

A visual overview of holographic node representations is in Figure 2. The expansion map introduces a dimension $T$, yielding node-level representations $V(\boldsymbol{A}, \boldsymbol{V}_1)_v \in \mathbb{R}^{T \times d_e}$ for each node $v$, instead of the typical "flat" embeddings in $\mathbb{R}^{d_e}$. This $T$ dimension encodes positional information (Property (2)) through $T$ distinct "views" of the graph, as shown by different colored vectors in Figure 2, addressing the impossibility result for "flat" embeddings in Proposition 2.3.

Whilst expressive, the expansion map alone does not give suitable node representations since information is distributed along the $T$ dimension, and does not respect any permutation symmetries. Property (1) ensures that the reduction map $R$ combines the information across the $T$ views, to output structural representations for any set of $r$ nodes. Since the reduction map depends on $r$ it is order specific ($r = 1$ for node, $r = 2$ for link, and so on), and it is learned anew for each task we aim to adapt to. The partition $\Lambda$ is crucial in this process, as it specifies how the $T$ representations should be grouped to obtain suitable structural representations.

While any expansion map $E$ can trivially lead to valid holographic node representations by choosing a constant reduction map (e.g., $R$ returning 0), the challenge lies in designing a reduction map that generates *expressive* representations. We address this in Section 4, proving the expressivity of our approach by leveraging the link between positional and structural representations, building on the existence result from Srinivasan & Ribeiro (2020). A key contribution of our work is advancing this theoretical result by designing practical architectures that efficiently perform this operation.

## 4 AN ARCHITECTURE FOR HOLOGRAPHIC NODE REPRESENTATIONS

This section presents HoloGNN, an architecture for learning holographic node representations. We begin by outlining how the symmetry-free expansion map is obtained through symmetry breakings, achieved by perturbing node features for selected nodes. Then, we propose two breaking methods. In Section 4.2, we describe *sequential breaking*, a general design inspired by numerical linear algebra methods, such as the Lanczos algorithm for computing eigenvectors (Lanczos, 1950). We formalize this connection, showing that sequential breaking generalizes eigenvector methods in a learnable and canonical way. In Section 4.3, we propose a faster, special case called *parallel breaking*.

## 4.1 OVERVIEW OF SYMMETRY BREAKINGS

The intuition behind HoloGNN is to obtain the symmetry-free expansion map $E$ through "symmetry breakings". This process breaks isomorphisms between nodes by perturbing the features of selected nodes, and then passing these modified features through a structural model. The perturbed features act as node identifiers, enabling the model to learn distinct representations for isomorphic nodes, which a structural model cannot do.

The idea of injecting unique features to obtain positional representations is well-established in previous work (You et al., 2019; Murphy et al., 2019; Abboud et al., 2021; Sato et al., 2021; Puny et al., 2020; Eliasof et al., 2023). HoloGNN distinguishes itself by carefully injecting identifiers in a way that a reduction map can later be designed to reintroduce order-specific permutation symmetries.

Precisely, step $t = 1$ initializes $\boldsymbol{V}_1$ as in Equation (1). At step $t > 1$, new node representations $\boldsymbol{V}_t \in \mathbb{R}^{n \times d}$ are produced by injecting a unique identifier, such as the standard basis vector $\mathbb{1}_v \in \mathbb{R}^n$, for the *breaking node* $v$, into a structural model $f_t$ (e.g., a permutation equivariant GNN). The role of the breaking node $v$ is to obtain an embedding for $v$ that distinguishes it from other nodes, which structural models cannot achieve. The breaking nodes are selected based on predefined functions known as node-breaking selectors, which deterministically return lists of nodes. As we will see, the partition $\Lambda$ is uniquely determined by the breaking nodes, and the final structural representations are produced by combining all $\boldsymbol{V}_t$ generated by breaking nodes returned by the same breaking selector.

The node-breaking selectors are defined as $(h_\lambda)_{\lambda=1}^L$. Each selector $h_\lambda$ returns a list of breaking nodes and must satisfy several properties:

**Definition 4.1** (Breaking selector $h_\lambda$). For $\lambda = 1, \ldots, L$,

$$(v_{\lambda,1}, \ldots, v_{\lambda,k_\lambda}) = h_\lambda(\boldsymbol{A}, \boldsymbol{V}_1) \tag{4}$$

denote the $\lambda$-th list of breaking nodes, sorted by node ID. The selector $h_\lambda$ is permutation-equivariant, meaning that if a node $v$ is in $h_\lambda(\boldsymbol{A}, \boldsymbol{V}_1)$, then all nodes isomorphic to $v$ in $G = (\boldsymbol{A}, \boldsymbol{V}_1)$ are also included. Additionally, $(h_\lambda)_{\lambda=1}^L$ are disjoint, ensuring that no node is included in more than one list.

The breaking selectors can be either pre-defined or learnable functions of the input graphs. Valid choices include having each $h_\lambda$ return all nodes with the same degree, or expressive choices where each $h_\lambda$ returns a list of all (and only) isomorphic nodes. Appendix A.1 expands on practical choices.

**Validity of Symmetry Breakings.** It may seem unclear if symmetry breaking generates $E(\boldsymbol{A}, \boldsymbol{V}_1)$ satisfying Definition 3.1. In particular, one might question whether perturbing features for a *single* node $v$ at a time is enough to produce different embeddings for all nodes in the graph. In principle it may be necessary to perturb multiple nodes at once, which would require at least quadratic number of forward passes through the structural models. Fortunately, we prove that this is not the case, and symmetry breaking with single-node perturbations produces valid expansion maps.

**Theorem 4.2** (Sufficiency of single-node breakings for holographic representations). *For any $G = (\boldsymbol{A}, \boldsymbol{X})$ and isomorphic nodes $u, v \in G$, with $u \neq v$ and having different neighborhoods, there exists $E$ obtained by single-node symmetry breakings such that $V(\boldsymbol{A}, \boldsymbol{V}_1)_v \neq V(\boldsymbol{A}, \boldsymbol{V}_1)_u$.*

## 4.2 SEQUENTIAL BREAKING ALGORITHM

The *sequential breaking* algorithm introduces symmetry-breaking perturbations iteratively, where each new symmetry is broken on the output of previous symmetry breakings. At each step $t > 1$, new node embeddings $\boldsymbol{V}_t \in \mathbb{R}^{n \times d_e}$ are generated by a structural model $f_t$ which takes as input: (1) the adjacency $\boldsymbol{A}$, (2) the embeddings produced in earlier steps $\boldsymbol{V}_{t-1}, \ldots, \boldsymbol{V}_1$, and (3) a perturbation $\mathbb{1}_v$ on the breaking node $v$. The breaking node $v$ is the next node in the sequence returned by the current breaking selector $h_\lambda$, with the index $\lambda$ incremented when the end of the list is reached. The pseudocode is detailed in Algorithm 1.

---
**Algorithm 1** Sequential Breaking

1:  $\Lambda_1 = 1$
2:  $\lambda = 2$
3:  $t = 2$
4:  **while** $t \leq T^\star$ **do**
5:      **for** $v \in h_\lambda(\boldsymbol{A}, \boldsymbol{V}_1)$ **do**
6:          $\boldsymbol{V}_t = f_t(\boldsymbol{A}, \boldsymbol{V}_{t-1}, \ldots, \boldsymbol{V}_1, \mathbb{1}_v)$
7:          $t = t + 1$
8:          $\Lambda_t = \lambda$
9:      **end for**
10:     $\lambda = \lambda + 1$
11: **end while**
12: $T = t$

---

The algorithm runs until a specified maximum number of iterations, $T^\star$. However, the total number of steps $T$ may exceed $T^\star$ to ensure that the current breaking selector's list is fully exhausted. This guarantees that all nodes having the same role, and therefore in the same list, are used as breaking nodes, preventing any improper selection that could differ for isomorphic graphs, and hinder the reduction map from constructing structural representations. The expansion map is defined as:

$$E(\boldsymbol{A}, \boldsymbol{V}_1) = \left( \text{Concat}(\boldsymbol{V}_T, \ldots, \boldsymbol{V}_1), (h_\lambda(\boldsymbol{A}, \boldsymbol{V}_1))_{\lambda=1}^L, \Lambda \right) \in \mathbb{R}^{n \times T \times d_e} \times (2^{[n]})^L \times \mathbb{N}^T, \tag{5}$$

with Concat concatenating along a new dimension, and $(h_\lambda(\boldsymbol{A}, \boldsymbol{V}_1))_{\lambda=1}^L$ and $\Lambda$ saved for use in the reduction map. The reduction map uses this information to appropriately combine the embeddings $\boldsymbol{V}_t$ to reintroduce permutation symmetries, adapting the representations to the task order at hand.

**Expressivity of sequential breaking.** Algorithm 1 describes a very general class of representations, as evidenced by several well-known models that are special cases. First, all structural models, such as GIN (Xu et al., 2019) and GCN (Kipf & Welling, 2017), are special cases by setting $f_1 = $ GIN/GCN, $f_t = $ Identity for $t > 1$, and ignoring breaking nodes entirely. Another important special case, which we prove next, is that sequential breaking can compute eigenvectors (and functions thereof). Even more importantly, sequential breaking resolves sign and basis ambiguities inherent in eigenvectors, eliminating the need for additional techniques like sign flipping (Dwivedi et al., 2023; Kreuzer et al., 2021; Kim et al., 2022; Dwivedi et al., 2021) or invariant (Wang et al., 2022; Lim et al., 2022; Huang et al., 2024b) and equivariant (Lim et al., 2023) networks to account for them.

**Theorem 4.3** (Expansion Map can express canonical eigenvectors). *For any symmetric matrix $\boldsymbol{A}$, there exists an expansion map $E(\boldsymbol{A}, \boldsymbol{V}_1)$ obtained by sequential breaking which can express eigenvectors with no sign or basis ambiguity, and identical for isomorphic graphs up to permutations.*

The proof is constructive, manually instantiating expansion maps that exactly correspond to popular algorithms for computing eigenvectors, such as the Lanczos algorithm. Because the proof is constructive, we are also able to prove that sequential breaking removes sign/basis ambiguities, which has practical advantages over explicit canonicalization (Kaba et al., 2023; Ma et al., 2023; 2024).

Having constructed an expressive expansion map, we are left to design appropriate reduction maps. Given two isomorphic graphs $G_1$ and $G_2$, there exists a permutation matrix $\boldsymbol{B}^{G_1 \to G_2}$ such that, $\boldsymbol{V}_t^{G_2} = \boldsymbol{B}^{G_1 \to G_2} \boldsymbol{V}_t^{G_1}, \forall t \in [T]$ (*c.f.*, Remark B.8). A natural way to construct structural representations of order $r$ is to treat the representations of the $r$ nodes across these isomorphic graphs as part of a set. Specifically, the structural representation of $\{v_1, \ldots, v_r\}$ is obtained by first computing an intermediate representation at each step $t$, using a learnable set function $\phi_t$, which takes as input the representations of $\{v_1, \ldots, v_r\}$ at step $t$ in graph $G_k$, i.e.,

$$\boldsymbol{U}_{k,t} = \phi_t\Big(\Big\{\big(\boldsymbol{B}^{G_1 \to G_k}\boldsymbol{V}_t\big)_{\pi^{G_k} \circ u}\Big\}_{\forall u \in \{v_1, \ldots, v_r\}}\Big). \tag{6}$$

Then, another learnable set function $\rho_t$ aggregates the intermediate representations $\boldsymbol{U}_{k,t}$ for all $G_k$,

$$\boldsymbol{U}_{t,\{v_1, \ldots, v_r\}} = \rho_t\Big(\big\{\boldsymbol{U}_{k,t}\big\}_{\forall k \in n!}\Big). \tag{7}$$

Finally, the reduction map can be obtained as the list of such representations for each $t$,

$$R(E(\boldsymbol{A}, \boldsymbol{V}_1))_{\{v_1, \ldots, v_r\}} = \Big(\boldsymbol{U}_{t,\{v_1, \ldots, v_r\}}\Big)_{t \in [T]}. \tag{8}$$

A key limitation of this approach is that it requires iterating over all possible $n!$ isomorphic graphs to construct $\boldsymbol{B}^{G_1 \to G_k}$. Appendix A.2 discusses practical approaches that use the breaking selectors to construct structural representations of order $r$, instead of iterating over isomorphic graphs.

## 4.3 Parallel Breaking Algorithm

**Algorithm 2** Parallel Breaking

1: $\Lambda_1 = \lambda$
2: $t = 2$
3: $\lambda = 2$
4: **while** $t \leq T^\star$ **do**
5:     **for** $v \in h_\lambda(\boldsymbol{A}, \boldsymbol{V}_1)$ **do**
6:         $\boldsymbol{V}_t = f_\lambda(\boldsymbol{A}, \boldsymbol{V}_1 \oplus \mathbb{1}_v)$
7:         $t = t + 1$
8:         $\Lambda_t = \lambda$
9:     **end for**
10:    $\lambda = \lambda + 1$
11: **end while**
12: $T = t$

In practice, breaking symmetries sequentially can be slow. To address this, we introduce parallel breaking, a faster variant of the sequential approach. Parallel breaking generates embeddings $\boldsymbol{V}_t$ using a structural model $f_\lambda$ that, in addition to $\boldsymbol{A}$, takes only a perturbed version of $\boldsymbol{V}_1$ for all $t$, rather than all prior embeddings $\boldsymbol{V}_{t-1}, \ldots, \boldsymbol{V}_1$. In this case, the embeddings $\boldsymbol{V}_t$ for different $t$ are only differentiated by breaking different nodes, with perturbations obtained as $\boldsymbol{V}_1 \oplus \mathbb{1}_v$, where $\oplus$ denotes concatenation. We remark here that: (1) the structural model $f_\lambda$ is the same for all breaking nodes belonging to the same $h_\lambda$ (as opposed to differing for each $t$ in

the sequential breaking, because it does not see previous breakings); (2) since there is no dependency on the step $t$, all $\boldsymbol{V}_t$ can be computed in parallel. The pseudocode is shown in Algorithm 2.

Identically to sequential breaking, the maximum number of iterations is $T^\star$, but the total number of steps $T$ may exceed $T^\star$ to ensure that the current breaking selector's list $h_\lambda(\boldsymbol{A}, \boldsymbol{V}_1)$ is exhausted. The expansion map is defined by concatenating the embeddings $\boldsymbol{V}_T, \ldots, \boldsymbol{V}_1$ along a new dimension, while also collecting the breaking selectors $(h_\lambda(\boldsymbol{A}, \boldsymbol{V}_1))_{\lambda=1}^L$ and the partition $\Lambda$ (c.f., Equation (5)).

For parallel breaking, the reduction map can be constructed in a simpler way, by aggregating representations $\boldsymbol{V}_t$ obtained from breaking nodes returned by the same $h_\lambda$. Specifically, for the structural representation of $\{v_1, \ldots, v_r\}$ (i.e., of order $r$), we first compute an intermediate representation for each step $t$, using a learnable set function $\phi_\lambda$, taking the representations of $\{v_1, \ldots, v_r\}$ at step $t$,

$$\boldsymbol{U}_{\lambda,t} = \phi_\lambda\Big(\big\{\boldsymbol{V}_{t,u}\big\}_{\forall u \in \{v_1,\ldots,v_r\}}\Big). \tag{9}$$

Then, another learnable set function $\rho_\lambda$ aggregates the intermediate representations $\boldsymbol{U}_{\lambda,t}$ for all $t$ obtained by breaking nodes returned by the same breaking selector, and therefore such that $\Lambda_t = \lambda$,

$$\boldsymbol{U}_{\lambda,\{v_1,\ldots,v_r\}} = \rho_\lambda\Big(\big\{\boldsymbol{U}_{\lambda,t}\big\}_{\forall t \in [T] \text{ s.t. } \Lambda_t = \lambda}\Big). \tag{10}$$

Finally, the reduction map can be obtained as the list of such representations for each $\lambda$,

$$R(E(\boldsymbol{A}, \boldsymbol{V}_1))_{\{v_1,\ldots,v_r\}} = \Big(\boldsymbol{U}_{\lambda,\{v_1,\ldots,v_r\}}\Big)_{\lambda \in [L]}. \tag{11}$$

The subtlety of this formulation is in which groups of representations the reduction map must be invariant to the ordering, and which groups it is beneficial to be sensitive to the order. Specifically, the within-breaking-set representations being order-invariant (Equation (10)) is essential for $R \circ E$ to be structural. Distinguishing the ordering of between-breaking-set representations (Equation (11)) is necessary for the representations to be *expressive*. Next, we show that these choices return holographic node representations, and are maximally expressive in the sense of Definition 2.2.

### 4.4 HoloGNN Gives Expressive Holographic Node Representations

In the previous subsections, we introduced two designs for expansion and reduction maps: sequential breaking (Section 4.2) and parallel breaking (Section 4.3). Sequential breaking enables the computation of canonical eigenvectors, while parallel breaking provides a more scalable solution. Importantly, both choices define holographic node representations, as the next result shows.

**Theorem 4.4** (Sufficiency of single-node symmetry breakings for structural of any order). *Let $R$ be the reduction map as defined in Equations (8) and (11). When combined with its corresponding expansion map $E$, then $R(E(\boldsymbol{A}, \boldsymbol{V}_1))_{\{v_1,\ldots,v_r\}}$ is a structural representation of $\{v_1, \ldots, v_r\}$.*

The significance of Theorem 4.4 lies in the sufficiency of single-node symmetry breakings, which are highly efficient, to generate structural representations. Moreover, we show that these breakings produce most-expressive link structural representations (per Definition 2.2) for the parallel breaking algorithm, with a corresponding result for the sequential breaking algorithm in Appendix B.3.

**Theorem 4.5** (Sufficiency of single-node breakings for most-expressive link structural). *Let $E$ be the parallel breaking expansion map in Equation (5) with $T^\star = n$ and let $R$ be the reduction map as defined in Equation (11) with $r = 2$. Assume $\rho_\lambda$ and $\phi_\lambda$ injective, and $f_\lambda$ node-most-expressive, for every $\lambda \in [L]$. Then, $R(E(\boldsymbol{A}, \boldsymbol{V}_1))_{\{v_1,v_2\}}$ is a most-expressive structural representation of $\{v_1, v_2\}$.*

While maximal expressivity can be achieved for links with single-node breakings, this result can be extended by considering multi-node symmetry breakings. Theorem B.4 in Appendix B.3 shows that breaking $r - 1$ nodes is sufficient to obtain most-expressive structural representations of order $r$.

## 5 Experiments

We empirically study the generalization of HoloGNN between tasks of different orders performed on the same dataset. We consider a typical pre-train-adapt protocol, where HoloGNN is pre-trained on a task of one order, e.g., link-prediction, and adapted to solve a new task of a potentially different

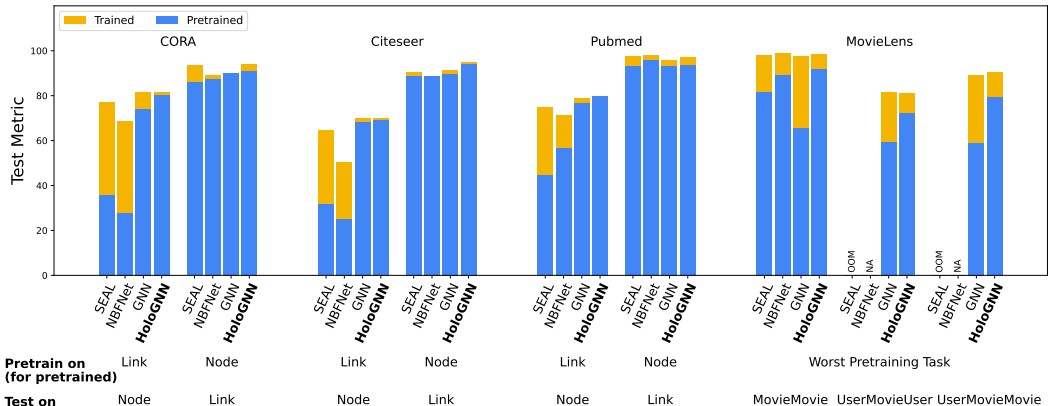

Figure 3: Performance (the higher the better) of each model when trained on one task (yellow bar) and when pre-trained on a different task and adapted to the task (blue bar). *For each model, the more yellow visible, the larger the performance loss from using pre-trained embeddings.* HoloGNN shows more consistent performance and thus smaller performance loss across all datasets and tasks.

order, e.g., node-classification. During adaptation, the expansion map remains *fixed*, with the pre-trained node embeddings retrieved from storage, while only the light-weight reduction map and a task-specific output head are trained on the new task. The protocol for other models is the same, with an MLP output head re-trained on the downstream test task on fixed pre-trained node embeddings.

We implement HoloGNN using the parallel breaking algorithm with number of symmetry breakings $T^\star \leq 10 \ll n$. Next we present our main results and refer to Appendix E for additional experiments.[1]

**Planetoid and MovieLens** (Sen et al., 2008; Fey & Lenssen, 2019). We compare HoloGNN against SEAL (Zhang & Chen, 2018; Li et al., 2020), NBFNet (Zhu et al., 2021), and a standard GNN with SAGE (Hamilton et al., 2017) or GCN (Kipf & Welling, 2017) layers, depending on the dataset. Notably, while both standard GNNs and HoloGNN produce pre-training node embeddings regardless of the pre-training task order, SEAL and NBFNet generate pre-training embeddings that match the order of the pre-training task, leading to significantly higher storage requirements for high-order tasks. Figure 3 (and Tables 3 and 4) shows the performance of each model when trained on a particular task (yellow bar) and when pre-trained on a different task and adapted to it (blue bar). *For each model, the more yellow visible, the larger the performance loss from using pre-trained embeddings.*

HoloGNN demonstrates significantly more consistent performance when compared to other methods. For instance, in the CORA dataset, the difference in performance between HoloGNN when trained on **Node Classification** (81.6%) and its performance when pre-trained on **Link Prediction** and adapted to **Node Classification** (80.2%) results in a performance drop of less than 1.5%. On the contrary, SAGE suffers from a loss of 7%, while NBFNet and SEAL show a more dramatic drop of 41%. Similarly, in the MovieLens dataset, when considering the order-3 task **UserMovieUser**, the difference in performance between HoloGNN pre-trained on the task (81.1%) and its worst performance when pre-trained in any other task (72.1%, obtained pre-training on the order-2 **MovieMovie**) results in a performance drop of 9%. In comparison, SAGE suffers a more pronounced 22% drop, SEAL runs out of memory (OOM), and NBFNet is unable to manage tasks of order-3 (NA).

**RelBench.** We evaluate HoloGNN on RelBench (Robinson et al., 2024), a recently proposed benchmark of graph datasets derived from relational databases. We use the `rel-stack` dataset, which is obtained from several years of activity on the Stack Exchange Q&A website. We evaluate HoloGNN on (i) two node level tasks: `user-engagement` and `user-badge`, where the task is to predict if a user will be active and if a user will earn a badge (community aware) in the next 3 months, and (ii) one link-level task: `user-post-comment`, where the goal is to predict if a user will comment on a given post in the next 3 months. Following Robinson et al. (2024) our implementation is based on SAGE (Hamilton et al., 2017), and we include task-agnostic non-learnable baselines.

Results are shown in Table 1. Notably, when pre-training on node-level tasks (`user-engagement` and `user-badge`) and adapting to link-level (`user-post-comment`), SAGE completely fails

---

[1]Our code is available at https://github.com/beabevi/holognn

Table 1: Evaluation on the `rel-stack` dataset from RelBench. The `user-post-comment` task is link prediction (order-2), and the other two are node classification (order-1) tasks. HoloGNN achieves superior performance when adapted to tasks of different order than the pre-training one. Results in blue indicate cases where pre-train and test tasks are the same (i.e., supervised learning).

| | | | TEST TASK | | |
|---|---|---|---|---|---|
| | | | user-post-comment (MAP ↑) | user-engagement (AUROC ↑) | user-badge (AUROC ↑) |
| TASK-AGNOSTIC | | Global Popularity | $0.03\pm0.00$ | NA | NA |
| | | Past Visit | $2.05\pm0.00$ | NA | NA |
| | | Random | NA | $50.7\pm0.20$ | $50.1\pm0.00$ |
| | | Majority | NA | $50.1\pm0.20$ | $50.0\pm0.00$ |
| PRE-TRAINING TASK | user-post-comment | SAGE | $0.11\pm0.05$ | $88.2\pm0.05$ | $84.6\pm0.05$ |
| | | HoloGNN | $3.40\pm0.29$ | $88.8\pm0.10$ | $85.5\pm0.00$ |
| | user-engagement | SAGE | $0.00\pm0.00$ | $90.7\pm0.01$ | $86.0\pm0.04$ |
| | | HoloGNN | $2.38\pm0.33$ | $90.6\pm0.01$ | $87.0\pm0.03$ |
| | user-badge | SAGE | $0.00\pm0.00$ | $89.2\pm0.01$ | $89.0\pm0.00$ |
| | | HoloGNN | $1.02\pm0.21$ | $89.4\pm0.05$ | $89.0\pm0.05$ |

Table 2: Link prediction AUC and runtime per epoch in the tasks of Lim et al. (2023). We show that eigenvectors obtained as output of the expansion map (Algorithm 1) results in comparable or better performance than the best performing Sign Equivariant, while being significantly faster.

| | Erdős-Rényi | | Barabási-Albert | |
|---|---|---|---|---|
| Model | Test AUC | Runtime (s) | Test AUC | Runtime (s) |
| GCN (constant input) | $.497\pm.06$ | $.058\pm.00$ | $.705\pm.01$ | $.048\pm.00$ |
| SignNet (Lim et al., 2022) | $.498\pm.00$ | $.120\pm.00$ | $.707\pm.00$ | $.095\pm.00$ |
| $\boldsymbol{U}_{i,:}^{\top}\boldsymbol{U}_{j,:}$ | $.570\pm.01$ | $.010\pm.01$ | $.597\pm.01$ | $.008\pm.00$ |
| $\mathrm{MLP}(\boldsymbol{U}_{i,:} \odot \boldsymbol{U}_{j,:})$ | $.614\pm.02$ | $.050\pm.00$ | $.651\pm.03$ | $.040\pm.00$ |
| Sign Equivariant (Lim et al., 2023) | $\mathbf{.751}\pm.00$ | $.063\pm.00$ | $\underline{.773}\pm.01$ | $.054\pm.00$ |
| Eigenvectors via Expansion Map | $\underline{.750}\pm.00$ | $.050\pm.00$ | $\mathbf{.816}\pm.00$ | $.040\pm.00$ |

to generalize, achieving 0.0 MAP. HoloGNN, on the other hand, successfully transfers from node-to link-level tasks. Similarly, when pre-training on the link-level and adapting to node-level tasks, HoloGNN suffers a smaller drop in performance than SAGE. For instance, when transferring to `user-badge` from `user-engagement`, SAGE suffers a $3\%$ drop in performance, whilst HoloGNN suffers only $2\%$. Note that the improved performance of HoloGNN is not simply due to being a stronger base model. When directly training on a task, HoloGNN achieves identical performance to SAGE. This shows that the key differentiator between them is in their transfer properties.

**Eigenvectors.** To demonstrate the generality of holographic node representations, we employ the sequential breaking algorithm (*c.f.*, Algorithm 1) to obtain eigenvectors, which we use in the link prediction task of Lim et al. (2022). Table 2 shows that the eigenvectors returned by holographic node representations outperform the best performing methods in a fraction of the time. Our approach surpasses the performance of eigenvectors obtained by conventional eigensolvers ($\mathrm{MLP}(\boldsymbol{U}_{i,:}\odot\boldsymbol{U}_{j,:})$, with $\boldsymbol{U}_{i,:}$ the eigenvector embedding of node $i$). We attribute this improvement to our eigenvectors being derived from $\boldsymbol{V}_1$ (Equation (1)) which is learned on the task. That is, holographic node representations use structural features to select the combination of sign and basis, resulting in representations that facilitate learning rather than relying on arbitrary choices of conventional eigensolvers.

## 6 CONCLUSION

In this work, we study the ability of node embeddings to solve different *task orders*, such as predictions about single nodes, pairs of nodes, or larger sets. Our first insight is that no existing model is able to solve different task orders simultaneously. We trace the source of this limitation back to the permutations symmetries possessed by different embeddings, which are task-order dependent, and therefore no single choice of symmetries suffices. To address this challenge, we propose *holographic node representations*, which learn symmetry-free node representations, but done carefully so that the appropriate task-order symmetries can be re-introduced when adapting to new task orders. We propose HoloGNN, the first practical model for learning holographic node representations, and show that HoloGNN leads to significant improvements when adapting to new task orders.

**Reproducibility Statement.** The assumptions for our theoretical results can be found in Appendix B. The experimental details are outlined in Appendix E. Detailed steps of the method are shown in Algorithms 1 and 2, with remarks in Appendix A.

ACKNOWLEDGMENTS

BR acknowledges support from the National Science Foundation (NSF) awards CCF-1918483, CA-REER IIS-1943364 and CNS-2212160, an Amazon Research Award, and AnalytiXIN, Wabash Heartland Innovation Network (WHIN), Ford, NVidia, CISCO, and Amazon. Computing infrastructure was supported in part by CNS-1925001 (CloudBank). This work was supported in part by AMD under the AMD HPC Fund program.

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

# A  ADDITIONAL REMARKS AND TECHNICAL DETAILS ON HOLOGRAPHIC NODE REPRESENTATIONS

In this section, we provide supplementary information, technical clarifications, and remarks on holographic node representations and its practical instantiation with HoloGNN. We start by discussing in detail the definition of holographic node representations (*c.f.*. Definition 3.1).

Properties (1) and (2) are essential to ensuring that holographic node representations can be trained in a task-order agnostic way, whilst still being able to map representations back to order-specific representations. Property (1) guarantees final task-specific representations are structural of order $r$; and Property (2) ensures that the output of the expansion map consists of $T$ positional, and thus symmetry-free, representations, which are necessary for task-agnostic node representations due to the impossibility result from Proposition 2.3.

We further remark that Property (1) in Definition 3.1 ensures that the reduction map returns structural representations for *sets rather than tuples* of $r$ nodes, aligning with the observation made by Srinivasan & Ribeiro (2020) that structural representations should be defined for sets of nodes. The rationale behind this is that it is not possible to distinguish between isomorphic nodes[2] that occupy different positions within a tuple, i.e. $(v_1, v_2, \dots v_r)$ and $(v_2, v_1, \dots v_r)$ if $v_1$ and $v_2$ are isomorphic. Therefore all tuples containing the same nodes should be considered as the same set.

Property (2) in Definition 3.1 ensures that the expansion map $E$ returns $T$ positional representations for each node, where each positional representation correspond to a "view" of the graph. Importantly, in the case of sequential breaking, these positional representations can be $T$ (canonical) eigenvectors of any symmetric matrix of the graph (Theorem 4.3). Notably, encompassing eigenvectors has an additional implication, which is made clear in Definition 3.1 (Property 2). Specifically, since returning eigenvectors of $\boldsymbol{A}\boldsymbol{A}^T$ and $\boldsymbol{A}^T\boldsymbol{A}$ represents a valid expansion map output, for any undirected graph, if $u$ and $v$, $u \neq v$, are isomorphic nodes *with the same neighborhood*, then $V(\boldsymbol{A}, \boldsymbol{V}_1)_v = V(\boldsymbol{A}, \boldsymbol{V}_1)_u$ (Cotta et al., 2023, Theorem 4). Therefore, we exclude this case by requiring Property (2) to hold for isomorphic nodes that do not have the same neighborhood.

While Definition 3.1 provides a general definition, HoloGNN represents a practical instantiation, which obtains the symmetry-free expansion map by breaking symmetries for selected nodes chosen by breaking selectors. Intuitively, the role of the node breaking selectors is to determine the breaking nodes to be used in the expansion map, ensuring that positional representations are constructed in a way that allows the reduction map to generate structural representations. In other words, the final structural representations are produced by combining positional representations generated by breaking nodes returned by the same breaking selector (see e.g., Equation (10)). This ensures that the final representations are the same for isomorphic node sets (and therefore they are structural). In contrast, randomly selecting breaking nodes without node breaking selectors would not ensure the same final representations for isomorphic sets, nor would it provide the same representations across different runs or isomorphic graphs.

Definition 4.1 outlines the properties of the breaking selectors, which are necessary for designing an appropriate reduction map capable of returning structural representations. Importantly, Definition 4.1 can be specialized to define most-expressive selectors, as we present next. These most-expressive selectors are important because are those employed in the sequential breaking algorithm for obtaining eigenvectors, as we shall see in Appendix B.2.

**Definition A.1** (Most-expressive breaking selector $h_\lambda^\star$)**.** A most-expressive selector is a selector, as defined in Definition 4.1, that is most-expressive based on the graph topology. That is, for any two nodes $u, v \in G = (\boldsymbol{A}, \boldsymbol{V}_1)$, then $u, v \in h_\lambda^\star(\boldsymbol{A}, \boldsymbol{V}_1) \iff u, u$ isomorphic in $G$ based only on the graph topology $\boldsymbol{A}$.

In other words, when using $h_\lambda^\star$ then $v_{\lambda,1}, \dots, v_{\lambda,k_\lambda^\star}$ are isomorphic nodes in $\boldsymbol{A}$. Importantly, each $h_\lambda$ returns a different list (a different group of isomorphic nodes). This is always possible as non-isomorphic nodes can be distinguished by structural properties (e.g., differing degrees) – as otherwise they would be isomorphic – which a most-expressive selector is able to capture by definition.

---

[2]Two nodes $u$ and $v$ are isomorphic if there exists a permutation of nodes that swaps $u$ and $v$ such that the permuted adjacency and feature matrices remain identical.

## A.1   NODE BREAKING SELECTORS

Having established that the expansion map produces highly expressive symmetry-free representations, we now specify how the breaking nodes are chosen. In principle *any* structural rule may be used to construct the breaking selectors (Definition 4.1), as long as the choice is appropriately coupled with the definition of $\Lambda$ to recover from these choices when constructing structural representations (Definition 3.1). In the following, we discuss the breaking rules we consider:

(1) **Node degree.** Each breaking selector $h_\lambda$ returns all nodes that share the same degree[3]. That is, we consider $h_1$ to return all nodes with the highest degree, $h_2$ to return nodes with the second-highest degree, and so forth (inverse sorting is also a possibility). Nodes returned by the same breaking selector are sorted by node ids, as per Definition 4.1. In this case, $\Lambda$ ensures that $\Lambda_t$ remains identical across all $t$ for breaking nodes with the same degree.

(2) **Learned structural distribution.** Given $\boldsymbol{V}_1$, define a node-wise score $\mathbf{s} = \boldsymbol{V}_1 \mathbf{w} \in \mathbb{R}^n$ with learnable weights $\mathbf{w} \in \mathbb{R}^d$. Then, sample $k \leq n$ nodes without replacement according to the distribution Softmax($\mathbf{s}$). For the learned distribution we need to compute gradients through the random sample. For this we use the straight-through Gumbel-Softmax estimator (Jang et al., 2017; Maddison et al., 2017), to allow differentiable sampling (of $k$ nodes) from the distribution. This rule results in a single breaking selector $h_1$ containing all $k$ sampled nodes. While this rule is not permutation-equivariant in general, it can still be useful for large graphs, where there usually are no isomorphic nodes. During inference, instead of sampling, we directly select the top $k$ nodes based on their scores. If there is a tie at the end (i.e., the $k$-th and the $k+1$-th nodes have the same score), we include all nodes with the tied score, resulting in a list of at least $k$ nodes returned by $h_1$. In this way, $h_1$ is permutation equivariant in test.

(3) **Node ids, when $T^\star = n$.** The simplest rule involves a single breaking selector, $h_1$, that returns all nodes, sorted by their IDs. In this case, $T^\star = n$ because the algorithm requires iterating through the entire list of nodes before termination, making this choice less scalable than others. Furthermore, $\Lambda_t = 1$ for all $t$, reflecting the uniform treatment of all nodes.

(4) **Most expressive.** Each breaking selector $h_\lambda$ returns a list of nodes that are isomorphic to each other (Definition A.1). Recall that each $h_\lambda$ must return a different list (a different group of isomorphic nodes), which is always possible as non-isomorphic nodes can be distinguished by structural properties. Practical choices include employing most-expressive GNNs, such as Murphy et al. (2019); Dasoulas et al. (2021), or also expressive GNNs whose expressivity level is sufficient to distinguish all graphs in the family of graphs considered in the tasks of interest. We refer the reader to Morris et al. (2023) for a review of expressive methods.

Finally, we remark here that there are many other valid rules for selecting breaking nodes, whose exploration we leave for future work.

## A.2   A PRACTICAL SEQUENTIAL BREAKING REDUCTION MAP

Section 4.2 in the main paper presents an instantiation of the reduction map for the sequential breaking algorithm, which leverages the representations of $\{v_1, \ldots, v_r\}$ in all graphs isomorphic to the input one. A key limitation of this approach is that it requires iterating over each possible isomorphic graph $G_k$ to construct $\boldsymbol{B}^{G_k}$, which is then used in Equation (6) to get the representation of $\{v_1, \ldots, v_r\}$ in $G_k$. An alternative strategy is to consider the representations of all $r$ sets isomorphic to $\{v_1, \ldots, v_r\}$, instead of the representations of $\{v_1, \ldots, v_r\}$ for all possible $G_k$. However, identifying all isomorphic sets can be computationally intensive. A reasonable compromise is to consider all sets of $r$ nodes where *each node* is isomorphic to a different node in $\{v_1, \ldots, v_r\}$. These include the sets isomorphic to $\{v_1, \ldots, v_r\}$, as well as those that are not, even if their nodes are in a one-to-one mapping.

Specifically, these sets can be obtained from our breaking selectors $(h_\lambda(\boldsymbol{A}, \boldsymbol{V}_1))_{\lambda=1}^L$. Each set $s_{v_i, w}$ is generated by replacing node $v_i$ in $\{v_1, \ldots, v_r\}$ with another node $w$ taken from the same breaking selector list, i.e.,

$$s_{v_i, w} = \{v_1, \ldots, v_{i-1}, w, v_{i-1}, \ldots v_r\}.$$

---

[3] Any other structural property can be used in this case, e.g., betweenness centrality.

where $v_i \in \{v_1, \ldots, v_r\}$ and $v_i, w \in h_\lambda(\boldsymbol{A}, \boldsymbol{V}_1)$ for some $\lambda \in [L]$. We denote the collection of these sets as $\mathcal{S}_{\{v_1, \ldots, v_r\}}$, that is,

$$\mathcal{S}_{\{v_1, \ldots, v_r\}} = \{s_{v_i, w}\}_{\forall v_i \in \{v_1, \ldots, v_r\}, \forall w \in G \text{ s.t. } w, v_i \in h_\lambda(\boldsymbol{A}, \boldsymbol{V}_1)}.$$

Since $(h_\lambda(\boldsymbol{A}, \boldsymbol{V}_1))_{\lambda=1}^L$ is structural by definition (*c.f.*, Definition 4.1), the above strategy allows us to construct the set of sets still containing all isomorphic sets to $\{v_1, \ldots, v_r\}$, although $\mathcal{S}_{\{v_1, \ldots, v_r\}}$ also contains more sets not isomorphic to $\{v_1, \ldots, v_r\}$. To obtain a structural representation of $\{v_1, \ldots, v_r\}$, we first obtain an intermediate representation for each set $s_{v_i, w}$,

$$\boldsymbol{U}_{\lambda, s_{v_i, w}} = \phi_\lambda \Big( \{\boldsymbol{V}_{t, u}\}_{\forall u \in s_{v_i, w}} \Big). \tag{12}$$

Then, we aggregate these intermediate representations for all sets in $\mathcal{S}_{\{v_1, \ldots, v_r\}}$, that is

$$\boldsymbol{U}_{\lambda, t} = \psi_\lambda \Big( \{\boldsymbol{U}_{\lambda, s_{v_i, w}}\}_{\forall s_{v_i, w} \in \mathcal{S}_{\{v_1, \ldots, v_r\}}} \Big). \tag{13}$$

Next, we aggregate these intermediate representations for all $t$ generated by breaking nodes returned by the same breaking selector,

$$\boldsymbol{U}_{\lambda, \{v_1, \ldots, v_r\}} = \rho_\lambda \Big( \{\boldsymbol{U}_{\lambda, t}\}_{\forall t \in [T] \text{ s.t. } \Lambda_t = \lambda} \Big). \tag{14}$$

Finally, the reduction map can then be expressed as

$$R(E(\boldsymbol{A}, \boldsymbol{X}))_{\{v_1, \ldots, v_r\}} = (\boldsymbol{U}_{\lambda, \{v_1, \ldots, v_r\}})_{\forall \lambda \in [L]}. \tag{15}$$

## B  PROOFS

This appendix includes the proofs for the theoretical results presented in Sections 2 and 4. We start by formally proving the impossibility of having a *single* embedding capable of solving different task orders, particularly node and link predictions. Then, we discuss the results related to the expansion map, including its connection to canonical eigenvectors. Lastly, we present the proofs concerning the reduction map.

### B.1  SECTION 2 PROOFS

We begin by formalizing the challenges related to node embeddings and their use for solving different tasks. Let $f(\boldsymbol{A}, \boldsymbol{X}) \in \mathbb{R}^{n \times d}$ denote node embeddings. We define a node-level task predictor and a link-level task predictor, respectively, as

$$\widehat{Y}(\boldsymbol{A}, \boldsymbol{X})_u = \text{MLP}_{\text{node}}(f(\boldsymbol{A}, \boldsymbol{X})_u),$$
$$\widehat{Y}(\boldsymbol{A}, \boldsymbol{X})_{uv} = \text{MLP}_{\text{link}}(f(\boldsymbol{A}, \boldsymbol{X})_u, f(\boldsymbol{A}, \boldsymbol{X})_v).$$

In practice, $\text{MLP}_{\text{node}}$ and $\text{MLP}_{\text{link}}$ correspond to different heads on top of the single embeddings $f(\boldsymbol{A}, \boldsymbol{X})$. This implies that $f(\boldsymbol{A}, \boldsymbol{X})$ is used for solving two different tasks, a node-level task, which we denote by $\mathcal{T}_{\text{node}}$, and link-level task, which we denote by $\mathcal{T}_{\text{link}}$. We define the respective test errors as:

$$\mathcal{L}_{\mathcal{T}_{\text{node}}}(\mathcal{D}_{\text{node}}) = \mathbb{E}_{(\boldsymbol{A}, \boldsymbol{X}), \boldsymbol{Y}_u \sim \mathcal{D}_{\text{node}}}[\ell_{\mathcal{T}_{\text{node}}}(\widehat{Y}(\boldsymbol{A}, \boldsymbol{X})_u, \boldsymbol{Y}_u)],$$
$$\mathcal{L}_{\mathcal{T}_{\text{link}}}(\mathcal{D}_{\text{link}}) = \mathbb{E}_{(\boldsymbol{A}, \boldsymbol{X}), \boldsymbol{Y}_{uv} \sim \mathcal{D}_{\text{link}}}[\ell_{\mathcal{T}_{\text{link}}}(\widehat{Y}(\boldsymbol{A}, \boldsymbol{X})_{uv}, \boldsymbol{Y}_{uv})],$$

with $\ell_{\mathcal{T}_{\text{node}}}, \ell_{\mathcal{T}_{\text{link}}}$ appropriate loss functions, and denoting the test minima as $\mathcal{L}^*_{\mathcal{T}_{\text{node}}}$ and $\mathcal{L}^*_{\mathcal{T}_{\text{link}}}$. In the following, we formally prove that $f(\boldsymbol{A}, \boldsymbol{X}) \in \mathbb{R}^{n \times d}$ is not sufficient to accurately solve both tasks.

**Proposition B.1** (Impossibility of accurate any-order task learning from node embeddings). *Consider simultaneously performing two tasks, $\mathcal{T}_{node}$ and $\mathcal{T}_{link}$, using node embeddings $f(\boldsymbol{A}, \boldsymbol{X}) \in \mathbb{R}^{n \times d}$. There exist $\mathcal{T}_{node}$ and $\mathcal{T}_{link}$ such that, for any $MLP_{node}$ and $MLP_{link}$ achieving the training minima, no $f$ that produces either positional or structural representations can simultaneously satisfy the following two conditions: (1) $\mathcal{L}_{\mathcal{T}_{node}}(\mathcal{D}_{node}) = \mathcal{L}^*_{\mathcal{T}_{node}}$; (2) $\mathcal{L}_{\mathcal{T}_{link}}(\mathcal{D}_{link}) = \mathcal{L}^*_{\mathcal{T}_{link}}$. That is, when using standard (flat) node embeddings, the predictions cannot be simultaneously accurate (in test) for both tasks.*

*Proof.* The proof proceeds by showing that, for any $f$, it is always possible to construct either $\mathcal{T}_{\text{node}}$ or $\mathcal{T}_{\text{link}}$, such that one of the conditions does not hold. Note that $f$ returns node embeddings, which can either be equivariant to node permutations, i.e., structural, or not, i.e., positional. In the following, we will prove the theorem for these two cases separately.

Assume that $f$ is equivariant to node permutations, and therefore structural. Then, by definition, $\widehat{Y}_u$ and $\widehat{Y}_{uv}$ are also invariant to node permutations. It follows from Srinivasan & Ribeiro (2020); Zhang et al. (2021) that there exists a link-level task $\mathcal{T}_{\text{link}}$ such that $\mathcal{L}_{\mathcal{T}_{\text{link}}}(\mathcal{D}_{\text{link}}) \neq \mathcal{L}^*_{\mathcal{T}_{\text{link}}}$. Specifically, consider the link-level task consisting in predicting in test the two non-isomorphic links $(v_1, v_2)$ and $(v_1, v_3)$ in Figure 1, and assume that those have different labels. Since $f$ is equivariant to node permutations, the isomorphic nodes $v_2$ and $v_3$ will have the same representation, and therefore $(v_1, v_2)$ and $(v_1, v_3)$ will receive the same prediction, despite having different labels. This implies that $\mathcal{L}_{\mathcal{T}_{\text{link}}}(\mathcal{D}_{\text{link}}) \neq \mathcal{L}^*_{\mathcal{T}_{\text{link}}}$, since the test error can be reduced with correct predictions for these two links.

Assume that $f$ is not equivariant to node permutations, and therefore positional. Consider a node-level task $\mathcal{T}_{\text{node}}$ on a training graph and consider a node $u$. Now, assume that the test graph is composed of two disconnected copies to the training graph. This means that there are two nodes isomorphic to $u$ in the test graph. However their representations will be different by definition of positional encoding. Since the representation seen in training for $u$ will correspond to only one of two representations seen in test, the representation of one of the test nodes may lead to incorrect predictions. Therefore $\mathcal{L}_{\mathcal{T}_{\text{node}}}(\mathcal{D}_{\text{node}}) \neq \mathcal{L}^*_{\mathcal{T}_{\text{node}}}$.

Therefore, we have proved that for any $f$ returning node embeddings, there exists a task such that one condition does not hold, which concludes our proof. $\square$

The above result justifies the need for our holographic node representations (Definition 3.1), as simply using node embeddings, coupled with different heads for different task orders, cannot result in accurate predictions for different task orders.

## B.2 SECTION 4 EXPANSION MAP PROOFS

In the following, we present the theoretical results related to the expansion map, as introduced in Section 4.

We start by proving that single-node breakings are sufficient for obtaining the expansion mappings within holographic node representations.

**Theorem 4.2** (Sufficiency of single-node breakings for holographic representations). *For any $G = (\boldsymbol{A}, \boldsymbol{X})$ and isomorphic nodes $u, v \in G$, with $u \neq v$ and having different neighborhoods, there exists $E$ obtained by single-node symmetry breakings such that $V(\boldsymbol{A}, \boldsymbol{V}_1)_v \neq V(\boldsymbol{A}, \boldsymbol{V}_1)_u$.*

*Proof.* To prove the sufficiency of single-node breakings for a valid expansion map, we consider $E$ to be the expansion map of the sequential breaking algorithm. Since the parallel breaking algorithm represents a special case of the sequential algorithm, the result will naturally extend to the parallel case as well.

By assumption, $u$ and $v$ have different neighborhoods, and therefore there exists at least one node $w \in G$ that is neighbor of $u$ but not of $v$ (or vice-versa). Additionally, since the breaking selectors are non-overlapping (Definition 4.1) and assuming $T^\star = n$, there exists a step $t$ such that the breaking node is $w$. Then, it is sufficient to consider the structurally invariant model $f_t$ to perform one message passing step (with identity weights) using the perturbation $\mathbb{1}_w$, in which case $u$ will obtain a different representation than $v$, as $u$ is connected to $w$, that had perturbed features. Therefore, there exists a $t$ in which $f_t(\boldsymbol{A}, \boldsymbol{V}_{t-1}, \ldots, \boldsymbol{V}_1, \mathbb{1}_w)_u \neq f_t(\boldsymbol{A}, \boldsymbol{V}_{t-1}, \ldots, \boldsymbol{V}_1, \mathbb{1}_w)_v$. Since $E$ is obtained by concatenating the outputs of the different $f_t$ (Equation (5)), then $V(\boldsymbol{A}, \boldsymbol{V}_1)_v \neq V(\boldsymbol{A}, \boldsymbol{V}_1)_u$, which concludes our proof. $\square$

Next, we discuss one of the main results of this section, which is to show that there exists a sequential-breaking expansion map that can express eigenvectors with no sign or basis ambiguities. We consider symmetric matrices, which includes the normalized Laplacian matrix and the undirected adjacency matrix.

**Theorem 4.3** (Expansion Map can express canonical eigenvectors). *For any symmetric matrix $\boldsymbol{A}$, there exists an expansion map $E(\boldsymbol{A}, \boldsymbol{V}_1)$ obtained by sequential breaking which can express eigenvectors with no sign or basis ambiguity, and identical for isomorphic graphs up to permutations.*

We prove this result by constructing an explicit expansion map with $T^* = n$ for which the embeddings $\boldsymbol{V}_1, \ldots, \boldsymbol{V}_n \in \mathbb{R}^{n \times 1}$ correspond to full set of $n$ *Lanczos vectors* of the symmetric matrix $\boldsymbol{A}$ (which can be the Laplacian). We then show how these Lanczos vectors can be used to obtain the full set of $n$ eigenvectors. Crucially, we demonstrate that both the Lanczos vectors and the eigenvectors are free of sign and basis ambiguities and are permutation equivariant.

It is important to note that, while the expansion map may return $\boldsymbol{V}_1, \ldots, \boldsymbol{V}_{n+1}$, the set of Lanczos vectors corresponds to $\boldsymbol{V}_1, \ldots, \boldsymbol{V}_n$. This is because the initial $\boldsymbol{V}_1$ is also one of the vectors (although obtained without breaking any node), and $\boldsymbol{V}_{n+1}$ will be the zero vector (as there are only $n$ orthogonal vectors in $\mathbb{R}^n$).

A key intermediate step is to prove that sequential breaking can express the Lanczos algorithm, which computes the Lanczos vectors. We then demonstrate how to derive the symmetric tridiagonal matrix, that is one of the terms of the Lanczos algorithm, from the Lanczos vectors and the original matrix $\boldsymbol{A}$ using a simple permutation-equivariant function. This step is crucial, as the tridiagonal matrix shares the same eigenvalues as the original matrix $\boldsymbol{A}$, and the eigenvectors of the original matrix can be obtained from those of the tridiagonal matrix through a simple matrix multiplication. Once the tridiagonal matrix is constructed, many efficient eigensolvers can be applied to produce its eigenvectors, and consequently those of the original matrix.

Before stating the intermediate Lanczos result, we first introduce Lanczos' algorithm in detail.

### B.2.1 LANCZOS ALGORITHM

Lanczos Algorithm takes as input a symmetric matrix $\boldsymbol{A} \in \mathbb{R}^{n \times n}$, and returns a matrix $\boldsymbol{V}$ of orthogonal columns (the Lanczos vectors) and a symmetric tridiagonal matrix $\boldsymbol{T}$ with the same eigenvalues as $\boldsymbol{A}$. Lanczos has four main steps:

1. Select an initial arbitrary vector $\mathbf{v}_1 \in \mathbb{R}^n$ with unit Euclidean norm.

2. Initial iteration:
   (a) Let $\mathbf{w}'_1 = A\mathbf{v}_1$.
   (b) Let $\alpha_1 = \mathbf{w}'^T_1 \mathbf{v}_1$.
   (c) Let $\mathbf{w}_1 = \mathbf{w}'_1 - \alpha_1 \mathbf{v}_1$.

3. For $t = 2, \ldots, n$ do:
   (a) Let $\beta_t = \|\mathbf{w}_{t-1}\|$ (also Euclidean norm).
   (b) If $\beta_t \neq 0$, then let $\mathbf{v}_t = \mathbf{w}_{t-1}/\beta_t$,
      else pick as $\mathbf{v}_t$ an arbitrary vector with Euclidean norm 1 that is orthogonal to all of $\mathbf{v}_1, \ldots, \mathbf{v}_{t-1}$.
   (c) Let $\mathbf{w}'_t = A\mathbf{v}_t$.
   (d) Let $\alpha_t = \mathbf{w}^*_t \mathbf{v}_t$.
   (e) Let $\mathbf{w}_t = \mathbf{w}'_t - \alpha_t \mathbf{v}_t - \beta_t \mathbf{v}_{t-1}$.

4. Let $\boldsymbol{V}$ be the matrix with columns $\mathbf{v}_1, \ldots, \mathbf{v}_n$. Let

$$
\boldsymbol{T} = \begin{pmatrix}
\alpha_1 & \beta_2 & & & & & 0 \\
\beta_2 & \alpha_2 & \beta_3 & & & & \\
 & \beta_3 & \alpha_3 & \ddots & & & \\
 & & \ddots & \ddots & \beta_{n-1} & & \\
 & & & \beta_{n-1} & \alpha_{n-1} & \beta_n \\
0 & & & & \beta_n & \alpha_n
\end{pmatrix}
$$

Note that it is easy to obtain the tridiagonal matrix $\boldsymbol{T}$ from the matrix of Lanczos vectors $\boldsymbol{V}$, as $\boldsymbol{T} = \boldsymbol{V}^T \boldsymbol{A} \boldsymbol{V}$.

Because it is easy to obtain the tridiagonal matrix $T$ (from which it is easy to obtain eigenvectors, see Cuppen (1980)) from the Lanczos vectors, the precise statement of our intermediate result is that sequential breaking can return the Lanczos vectors. The proof of the full result (Theorem 4.3) then simply extends this by constructing a function of the Lanczos vectors and the matrix $A$ to additionally compute $T$, compute the eigenvectors of $T$, and derive those of $A$. We will show that all these steps are free from sign and basis ambiguities, and additionally prove that the Lanczos vectors (and consequently the tridiagonal and the eigenvectors) are permutation equivariant.

**Theorem B.2** (Sequential breaking can implement Lanczos Algorithm). *For any symmetric matrix $A \in \{0,1\}^{n \times n}$ there exists an $n$-step expansion map $E(A, X) = [V_n, \ldots, V_1]$, where $V_j$ is the $j$th Lanczos vector of $A$.*

*Proof.* To prove that the sequential breaking algorithm can implement the Lanczos algorithm, we need to define the functions $(h_\lambda)_{\lambda=1}^L$ and $f_t$ for $t \in [n]$ in a way that matches the steps of the Lanczos algorithm. Let's break this down step by step:

For $(h_\lambda)_{\lambda=1}^L$ we consider most-expressive breaking selectors (Definition A.1), which partition the set of nodes into isomorphism equivalence classes. We denote the breaking node of the $t$-step as $v_t$.

Additionally, we take the embedding dimension $d = 1$, as we want each $V_t$ to correspond to a Lanczos *vector*.

Now, we are ready to define $V_t = f_t(A, V_{t-1}, \ldots, V_1, \mathbb{1}_v)$, as follows:

1. For $t = 1$: $V_1 = g_{\text{struc}}(A, X) \in \mathbb{R}^n$, the (normalized) vector obtained from a structural function (Equation (1)). In this particular case, we assume that $V_1$ has norm 1 (as required by the Lanczos algorithm), and we further assume that $g_{\text{struc}}$ discards node features $X$, because eigenvector methods do not use node features but only the matrix $A$.

2. For $t = 2$:

   (a) $\mathbf{w}_1' = A V_1$
   (b) $\alpha_1 = V_1^T \mathbf{w}_1'$
   (c) $\mathbf{w}_1 = \mathbf{w}_1' - \alpha_1 V_1$
   (d) $\beta_2 = \|\mathbf{w}_1\|$
   (e) If $\beta_2 \neq 0$: $V_2 = \frac{\mathbf{w}_1}{\beta_2}$
   (f) Else: $V_2 = \frac{\mathbb{1}_{v_1} - \text{proj}_{V_1}(\mathbb{1}_{v_1})}{\|\mathbb{1}_{v_1} - \text{proj}_{V_1}\|} \in \mathbb{R}^n$ where proj denotes the Gram-Schmidt projection operator of $\mathbb{1}_{v_t}$ onto the span of $V_1$. Since $V_1$ is a single vector, this is simply $V_2 = \frac{\mathbb{1}_{v_t} - (\mathbb{1}_{v_t}^\top V_1) V_1}{\mathbb{1}_{v_t} - (\mathbb{1}_{v_t}^\top V_1) V_1}$.

3. If $t > 2$:

   (a) $\mathbf{w}_{t-1}' = A V_{t-1}$
   (b) $\alpha_{t-1} = V_{t-1}^T \mathbf{w}_{t-1}'$
   (c) $\mathbf{w}_{t-1} = \mathbf{w}_{t-1}' - \alpha_{t-1} V_{t-1} - \beta_{t-1} V_{t-2}$
   (d) $\beta_t = \|\mathbf{w}_{t-1}\|$
   (e) If $\beta_t \neq 0$: $V_t = \frac{\mathbf{w}_{t-1}}{\beta_t}$
   (f) Else: $V_t = \frac{\mathbb{1}_{v_t} - \text{proj}_{V_1, \ldots V_{t-1}}(\mathbb{1}_{v_t})}{\|\mathbb{1}_{v_t} - \text{proj}_{V_1, \ldots V_{t-1}}(\mathbb{1}_{v_t})\|} \in \mathbb{R}^n$, the orthonormalization of $\mathbb{1}_{v_t}$ with respect to the span of $V_1, \ldots V_{t-1}$.

Note that in each case this definition of $f_t$ is a composition of the following permutation equivariant functions: linear transformations, dot products, norms, linear combinations, and conditional statements. Because of this, $f_t$ is also permutation equivariant as required in Section 4.2.

It is not at first clear that this choice of $f_t$ is well defined. The danger comes from the Gram-Schmidt orthogonal projection $\mathbb{1}_{v_t} - \text{proj}_{V_1, \ldots V_{t-1}}(\mathbb{1}_{v_t})$, which must be non-zero so that it can be normalized. For now let us assume that this $f_t$ is well defined in order to show that the rest of the proof goes through. Then, at the end of the proof we will return to this matter.

With these choices the sequential breaking algorithm (Algorithm 1) becomes step-by-step identical to the Lanczos algorithm, with a particular instantiation of the "arbitrary" breaking vectors, which in the sequential breaking algorithm are not random vectors as in Lanczos, but are instead defined as $\boldsymbol{V}_1 = g_{\text{struc}}(\boldsymbol{A}, \boldsymbol{X}) \in \mathbb{R}^n$ and as $\mathbb{1}_{v_t}$, with $v_t$ the breaking node.

This formulation of the sequential breaking algorithm matches the Lanczos algorithm step by step. The key points to note are:

1. The function $f_t$ is defined to perform the exact same operations as the Lanczos algorithm for each $t$.

2. When $\beta_t = 0$, we choose $\boldsymbol{V}_t = \frac{\mathbb{1}_{v_t} - \text{proj}_{\boldsymbol{V}_1, \ldots \boldsymbol{V}_{t-1}}(\mathbb{1}_{v_t})}{\|\mathbb{1}_{v_t} - \text{proj}_{\boldsymbol{V}_1, \ldots \boldsymbol{V}_{t-1}}(\mathbb{1}_{v_t})\|}$, which for appropriate choices of most-expressive breaking selectors is guaranteed to be a unit vector orthogonal to previous vectors due to the Gram-Schmidt orthonormalization projection map, as we show in Theorem B.6. This underscores that this choice of $f_t$ is well defined.

3. The output $\boldsymbol{V}$ is constructed in the same way as in the Lanczos algorithm.

This demonstrates that with the given definitions of $(h_\lambda)_{\lambda=1}^L$ and $f_t$ for $t \in [n]$, the sequential breaking algorithm exactly replicates the Lanczos algorithm. $\qquad\square$

We have seen that the sequential breaking algorithm can implement the Lanczos algorithm. Importantly, since all choices are deterministic, with both the initial vector $\boldsymbol{V}_1$ and the arbitrary vectors determined by the most-expressive breaking selectors and therefore by the graph structure, the obtained Lanczos vectors do not have any ambiguity, with both sign and basis exactly determined by $g_{\text{struc}}$ and $(h_\lambda)_{\lambda=1}^L$. Furthermore, Theorem B.7 shows that the Lanczos vectors are identical up to permutation for isomorphic graphs.

Given these results we are now ready to prove Theorem 4.3.

*Proof of Theorem 4.3.* Let $f_1, \ldots, f_n$ and $(h_\lambda)_{\lambda=1}^L$ be as in Theorem B.2. We construct an additional function $\Gamma$ that, given the original matrix $\boldsymbol{A}$ and the Lanczos vectors, returns the eigenvectors of $\boldsymbol{A}$,
$$\boldsymbol{U} = \Gamma(\boldsymbol{A}, \boldsymbol{V}) \in \mathbb{R}^{n \times n}$$
with $\boldsymbol{V} = [\boldsymbol{V}_1, \ldots, \boldsymbol{V}_n]$ and $\boldsymbol{U}$ denoting the full set of $n$ eigenvectors of $\boldsymbol{A}$.

Specifically we define $\Gamma$ to be factorized into two steps $\Gamma = \Gamma_2 \circ \Gamma_1$, corresponding to the two remaining steps required to compute eigenvectors from $\boldsymbol{A}$ and $\boldsymbol{V}_1, \ldots, \boldsymbol{V}_n$, which we have seen do not have sign/basis ambiguities and are identical up to permutations for isomorphic graphs.

First, $\Gamma_1$ represents the map $(\boldsymbol{A}, \boldsymbol{V}) \mapsto (\boldsymbol{T}, \boldsymbol{V}) \in \mathbb{R}^{n \times n}$, returning the tridiagonal matrix of the Lanczos algorithm, as well as the Lanczos vectors (this last output is included for later usage in $\Gamma_2$, but simply corresponds to the input). Note that this $\Gamma_1$ can be written as
$$\Gamma_1(\boldsymbol{A}, \boldsymbol{V}) = (\boldsymbol{V}^T \boldsymbol{A} \boldsymbol{V}, \boldsymbol{V})$$
since $\boldsymbol{T} = \boldsymbol{V}^T \boldsymbol{A} \boldsymbol{V}$. Since $\Gamma_1$ simply consists of two matrix multiplications, it is permutation equivariant.

Then, $\Gamma_2 : \mathbb{R}^{n \times n} \times \mathbb{R}^{n \times n} \to \mathbb{R}^{n \times n}$ can be any of the (fast) specialized eigensolvers for symmetric tridiagonal matrices, followed by a matrix multiplication to recover the eigenvectors of $\boldsymbol{A}$ from those of $\boldsymbol{T}$. For concreteness, we may take $\Gamma_2$ to first perform the divide-and-conquer algorithm proposed by Cuppen (1980), which computes the eigendecomposition of $\boldsymbol{T}$ in $\mathcal{O}(n^2)$ steps. Importantly, it was shown by Dhillon (1997) that this divide-and-conquer can be done *deterministically* using a "twisted factorization" (see Section 3.1 therein) to initialize, and so the resulting eigenvectors do not have sign/basis ambiguities. Then, given the eigenvectors $\boldsymbol{U}_{\text{Tri}}$ of $\boldsymbol{T}$, the eigenvectors $\boldsymbol{U}$ of $\boldsymbol{A}$ can be obtained as $\boldsymbol{U} = \boldsymbol{V} \boldsymbol{U}_{\text{Tri}}$. Importantly, since it only involves a decomposition (without ambiguities) and a matrix multiplication, $\Gamma_2$ is also permutation equivariant.

Thus, we have shown that it is possible to recover the eigenvectors of $\boldsymbol{A}$ from the Lanczos vectors. Since both the Lanczos vectors and the mappings used to obtain the eigenvectors are free of sign

and basis ambiguities, and are permutation equivariant, the resulting eigenvectors will also be free of these ambiguities and permutation equivariant, concluding our proof. □

### B.3 SECTION 4 REDUCTION MAP PROOFS

In the following, we present the theoretical results related to the reduction map, as introduced in Section 4. We start by showing that the reduction map designs introduced for the sequential and parallel breaking algorithms are valid, as they return structural representations of order $r$ (Property (1) in Definition 3.1).

**Theorem 4.4** (Sufficiency of single-node symmetry breakings for structural of any order). *Let $R$ be the reduction map as defined in Equations* (8) *and* (11)*. When combined with its corresponding expansion map $E$, then $R(E(\boldsymbol{A}, \boldsymbol{V}_1))_{\{v_1,\dots,v_r\}}$ is a structural representation of $\{v_1,\dots,v_r\}$.*

*Proof.* We will show this result for the sequential breaking algorithm and for the parallel breaking algorithm separately.

**Sequential Breaking.** Equation (8) trivially satisfies the definition of structural representation (Definition 2.1), as it returns the representation of the set $\{v_1,\dots,v_r\}$ by averaging its representations across all isomorphic graphs.

**Parallel Breaking.** We need to show that Equation (11) satisfies the definition of structural representation (Definition 2.1). Specifically, we show that if $\{v_1,\dots,v_r\} \sim \{u_1,\dots,u_r\}, \boldsymbol{A} \sim \boldsymbol{A}', \boldsymbol{V}_1 \sim \boldsymbol{V}_1'$ then $R(E(\boldsymbol{A}, \boldsymbol{V}_1))_{\{v_1,\dots,v_r\}} = R(E(\boldsymbol{A}', \boldsymbol{V}_1'))_{\{u_1,\dots,u_r\}}$. Since the output of $R$ is a concatenation of intermediate representations, it is sufficient to show that $\boldsymbol{U}_{\lambda,\{v_1,\dots,v_r\}} = \boldsymbol{U}'_{\lambda,\{u_1,\dots,u_r\}}$. This is trivially true because: (1) the breaking selectors are structural, and therefore breaking nodes belonging to the same selector in $(\boldsymbol{A}_1, \boldsymbol{V}_1)$ are isomorphic to breaking nodes belonging to the same selector in $(\boldsymbol{A}_1', \boldsymbol{V}_1')$, (2) $f_t$ for $t \in [T]$ is structural. Therefore the reduction map in Equation (11) returns structural representations. □

Note that while we have proven the above result for Equation (8) for the sequential breaking algorithm, it is easy to see that it extends also to Equation (15), by following the same proof steps used for parallel breaking. Indeed, Equation (15) similarly considers the concatenation of intermediate representations, each obtained by aggregating representations obtained by structural models and structural selectors.

Next, we prove that the representations returned by the reduction map are most-expressive for links. We start by showing that this is case for Equation (11).

**Theorem 4.5** (Sufficiency of single-node breakings for most-expressive link structural). *Let $E$ be the parallel breaking expansion map in Equation* (5) *with $T^\star = n$ and let $R$ be the reduction map as defined in Equation* (11) *with $r = 2$. Assume $\rho_\lambda$ and $\phi_\lambda$ injective, and $f_\lambda$ node-most-expressive, for every $\lambda \in [L]$. Then, $R(E(\boldsymbol{A}, \boldsymbol{V}_1))_{\{v_1,v_2\}}$ is a* most-expressive *structural representation of $\{v_1, v_2\}$.*

*Proof.* Let $\{u_1, u_2\}$ be another set of two nodes, which is different from $\{v_1, v_2\}$. We will show that $R(E(\boldsymbol{A}, \boldsymbol{V}_1))_{\{v_1,v_2\}} = R(E(\boldsymbol{A}, \boldsymbol{V}_1))_{\{u_1,u_2\}}$ iff $\{v_1, v_2\} \sim \{u_1, u_2\}$, therefore satisfying the definition of most-expressive structural representations (Definition 2.2).

Recall that, since $T^\star = n$ we have $T = n+1$ because we need to break all the $n$ nodes, and therefore the equations can be rewritten as

$$\boldsymbol{U}_{\lambda,t} = \phi_\lambda\Big(\{\boldsymbol{V}_{t,v_1}, \boldsymbol{V}_{t,v_2}\}\Big)$$

$$\boldsymbol{U}_{\lambda,\{v_1,v_2\}} = \rho_\lambda\Big(\{\boldsymbol{U}_{\lambda,t}\}_{\forall t \in [n+1] \text{ s.t. } \Lambda_t = \lambda}\Big)$$

$$R(E(\boldsymbol{A}, \boldsymbol{V}_1))_{\{v_1,v_2\}} = \Big(\boldsymbol{U}_{\lambda,\{v_1,v_2\}}\Big)_{\lambda \in [L]}.$$

For simplicity, let us further denote by $S_{t,\{v_1,v_2\}}$ the set of the representations of $\{v_1, v_2\}$ at time $t$, i.e., $S_{t,\{v_1,v_2\}} = \{\boldsymbol{V}_{t,v_1}, \boldsymbol{V}_{t,v_2}\}$.

($\Rightarrow$) We will show that $R(E(\boldsymbol{A}, \boldsymbol{V}_1))_{\{v_1,v_2\}} = R(E(\boldsymbol{A}, \boldsymbol{V}_1))_{\{u_1,u_2\}} \Rightarrow \{v_1, v_2\} \sim \{u_1, u_2\}$. Since $\rho_\lambda$ and $\phi_\lambda$ are injective, we have

$$R(E(\boldsymbol{A}, \boldsymbol{V}_1))_{\{v_1,v_2\}} = R(E(\boldsymbol{A}, \boldsymbol{V}_1))_{\{u_1,u_2\}}$$
$$\Rightarrow \forall \lambda \in [L] \quad \{S_{t,\{v_1,v_2\}}\}_{\forall t \in [n+1] \text{ s.t. } \Lambda_t = \lambda} = \{S_{t,\{v_1,v_2\}}\}_{\forall t \in [n+1] \text{ s.t. } \Lambda_t = \lambda}$$
$$\Rightarrow \forall j \in [n+1], \exists k \in [n+1] \quad \text{such that} \quad \{\boldsymbol{V}_{j,v_1}, \boldsymbol{V}_{j,v_2}\} = \{\boldsymbol{V}_{k,u_1}, \boldsymbol{V}_{k,u_2}\}$$

which means that $\forall j \in [n+1], \exists k \in [n+1]$ such that

$$\{f_\lambda(\boldsymbol{A}, \boldsymbol{V}_1 \oplus \mathbb{1}_{v_j})_{v_1}, f_\lambda(\boldsymbol{A}, \boldsymbol{V}_1 \oplus \mathbb{1}_{v_j})_{v_2}\} = \{f_\lambda(\boldsymbol{A}, \boldsymbol{V}_1 \oplus \mathbb{1}_{v_k})_{u_1}, f_\lambda(\boldsymbol{A}, \boldsymbol{V}_1 \oplus \mathbb{1}_{v_k})_{u_2}\}$$

Since the above means that $\forall j \in [n+1], \exists k \in [n+1]$ where the sets are the same, then without loss of generality we have that[4]

$$f_\lambda(\boldsymbol{A}, \boldsymbol{V}_1 \oplus \mathbb{1}_{v_j})_{v_1} = f_\lambda(\boldsymbol{A}, \boldsymbol{V}_1 \oplus \mathbb{1}_{v_k})_{u_1}$$
$$f_\lambda(\boldsymbol{A}, \boldsymbol{V}_1 \oplus \mathbb{1}_{v_j})_{v_2} = f_\lambda(\boldsymbol{A}, \boldsymbol{V}_1 \oplus \mathbb{1}_{v_k})_{u_2}$$

Consider the first equation. Since $f_\lambda$ is node-most-expressive, then $v_1 \sim u_1$, which implies $\exists \boldsymbol{P}$ permuting $v_j$ with $v_k$ and $v_1$ with $u_1$[5] such that $\boldsymbol{A} = \boldsymbol{P}\boldsymbol{A}\boldsymbol{P}^T$ and $\boldsymbol{V}_1 \oplus \mathbb{1}_{v_k} = \boldsymbol{P}(\boldsymbol{V}_1 \oplus \mathbb{1}_{v_j})$. This implies that $(v_j, v_1) \sim (v_k, u_1)$. Similarly, for the second equation we have that $(v_j, v_2) \sim (v_k, u_2)$. Consequently, it must be $(v_1, v_2) \sim (u_1, u_2)$ (cause if they were not, then either $(v_j, v_1) \not\sim (v_k, u_1)$ or $(v_j, v_2) \not\sim (v_k, u_2)$, reaching a contradiction).

($\Leftarrow$) We will show that $\{v_1, v_2\} \sim \{u_1, u_2\} \Rightarrow R(E(\boldsymbol{A}, \boldsymbol{V}_1))_{\{v_1,v_2\}} = R(E(\boldsymbol{A}, \boldsymbol{V}_1))_{\{u_1,u_2\}}$. Suppose by contradiction that $R(E(\boldsymbol{A}, \boldsymbol{V}_1))_{\{v_1,v_2\}} \neq R(E(\boldsymbol{A}, \boldsymbol{V}_1))_{\{u_1,u_2\}}$ which means that there exist $\lambda \in [L]$ such that $\boldsymbol{U}_{\lambda,\{v_1,v_2\}} \neq \boldsymbol{U}_{\lambda,\{u_1,u_2\}}$. Since $\phi_\lambda, \rho_\lambda$ are injective, this implies that $\exists j \in [n+1]$ such that[6]

$$f_\lambda(\boldsymbol{A}, \boldsymbol{V}_1 \oplus \mathbb{1}_{v_j})_{v_1} \neq f_\lambda(\boldsymbol{A}, \boldsymbol{V}_1 \oplus \mathbb{1}_{v_k})_{u_1} \quad \forall k \in [n+1] \quad \text{such that} \quad \Lambda_j = \Lambda_k = \lambda$$

However, since $\{v_1, v_2\} \sim \{u_1, u_2\}$, $\exists k \in [n+1]$ with $\Lambda_k = \lambda$ such that considering $\boldsymbol{P}$ permuting $v_j$ with $v_k$ and $v_1$ with $u_1$ is such that $\boldsymbol{A} = \boldsymbol{P}\boldsymbol{A}\boldsymbol{P}^T$ and $\boldsymbol{V}_1 \oplus \mathbb{1}_{v_k} = \boldsymbol{P}(\boldsymbol{V}_1 \oplus \mathbb{1}_{v_j})$. Since $f_\lambda$ is structural, then

$$f_\lambda(\boldsymbol{A}, \boldsymbol{V}_1 \oplus \mathbb{1}_{v_j})_{v_1} = f_\lambda(\boldsymbol{A}, \boldsymbol{V}_1 \oplus \mathbb{1}_{v_k})_{u_1}$$

reaching a contradiction. $\qquad\square$

Next, we show that maximal expressivity can also be achieved by Equation (8).

**Proposition B.3** (Sufficiency of single-node sequential breakings for most-expressive link structural). *Let $E$ be the sequential breaking expansion map in Equation (5) with $T^\star = n$ and let $R$ be the reduction map as defined in Equation (8) with $r = 2$. Assume $\rho_t$ and $\phi_t$ are injective, and $f_t$ defined as in Theorem B.2 (that is, returning the Lanczos vectors). Then, $R(E(\boldsymbol{A}, \boldsymbol{V}_1))_{\{v_1,v_2\}}$ is a most-expressive structural representation of $\{v_1, v_2\}$.*

*Proof.* From the proof of Theorem 4.4 we already know that $R(E(\boldsymbol{A}, \boldsymbol{V}_1))_{\{v_1,v_2\}}$ is structural. We now only need to show it is most expressive, that is $R(E(\boldsymbol{A}, \boldsymbol{V}_1))_{\{v_1,v_2\}} = R(E(\boldsymbol{A}', \boldsymbol{V}_1'))_{\{v_1',v_2'\}}$ $\Rightarrow \exists \pi \in \mathbb{S}_n \; \boldsymbol{A} = \pi \circ \boldsymbol{A}, \; \boldsymbol{V}_1 = \pi \circ \boldsymbol{V}_1$ and $\{v_1, v_2\} = \pi \circ \{v_1', v_2'\}$. This follows from Srinivasan & Ribeiro (2020, Theorem 2) as the Lanczos vectors are most expressive positional representations. $\qquad\square$

While maximal expressivity can be achieved only for links in the case of single-node breakings, for the parallel breaking algorithm case, it is possible to extend the result when considering multi-node breakings. Specifically, we next show that breaking $r - 1$ nodes at a time is sufficient to obtain a *most-expressive structural representation of order $r$*.

---

[4] An alternative equivalent relation would be to swap $u_1$ with $u_2$ in the equation.

[5] Note that $v_j$ can also be equal to $v_k$, and in such case $\boldsymbol{P}$ permutes only $v_1$ with $u_1$.

[6] We can alternatively swap $v_1$ with $v_2$ and/or $u_1$ with $u_2$, without loss of generality.

**Theorem B.4** (Sufficiency of $(r-1)$-node joint breakings for most-expressive $r$ structural). *Let $E$ be the parallel breaking expansion mapping obtained by perturbing $(r-1)$-nodes at a time, and let $T^\star = \binom{n}{r-1}$. Let $R$ be the reduction mapping as defined in Equation* (11). *Assume $\rho_\lambda$ and $\phi_\lambda$ injective, and $f_\lambda$ node-most-expressive, for every $\lambda \in [L]$. Then, $R(E(\boldsymbol{A}, \boldsymbol{V}_1))_{\{v_1, \ldots v_r\}}$ is a most-expressive structural representation of $\{v_1, \ldots v_r\}$.*

*Proof.* The proof is step-by-step identical to the proof of Theorem 4.5, with the only difference being that we break $r-1$ nodes at a time to obtain most-expressive structural representations of order $r$. Let $\{u_1, \ldots, u_r\}$ be another set of $r$ nodes, which is different from $\{v_1, \ldots, v_r\}$. We will show that $R(E(\boldsymbol{A}, \boldsymbol{V}_1))_{\{v_1, \ldots, v_r\}} = R(E(\boldsymbol{A}, \boldsymbol{V}_1))_{\{u_1, \ldots, u_r\}}$ iff $\{v_1, \ldots, v_r\} \sim \{u_1, \ldots, u_r\}$, therefore satisfying the definition of most-expressive structural representations (Definition 2.2).

Recall that, since $T^\star = \binom{n}{r-1}$ we have $T = \binom{n}{r-1} + 1$ because we need to break all the $\binom{n}{r-1}$ set of $r-1$ nodes, and therefore the equations can be rewritten as

$$\boldsymbol{U}_{\lambda, t} = \phi_\lambda \Big( \{\boldsymbol{V}_{t, v_1}, \ldots, \boldsymbol{V}_{t, v_r}\} \Big)$$

$$\boldsymbol{U}_{\lambda, \{v_1, v_2\}} = \rho_\lambda \Big( \{\boldsymbol{U}_{\lambda, t}\}_{\forall t \in [T] \text{ s.t. } \Lambda_t = \lambda} \Big)$$

$$R(E(\boldsymbol{A}, \boldsymbol{V}_1))_{\{v_1, \ldots, v_r\}} = \Big( \boldsymbol{U}_{\lambda, \{v_1, \ldots, v_r\}} \Big)_{\lambda \in [L]}.$$

For simplicity, let us further denote by $S_{t, \{v_1, \ldots, v_r\}}$ the set of the representations of $\{v_1, \ldots, v_r\}$ at time $t$, i.e., $S_{t, \{v_1, \ldots, v_r\}} = \{\boldsymbol{V}_{t, v_1}, \ldots, \boldsymbol{V}_{t, v_r}\}$.

($\Rightarrow$) We will show that $R(E(\boldsymbol{A}, \boldsymbol{V}_1))_{\{v_1, \ldots, v_r\}} = R(E(\boldsymbol{A}, \boldsymbol{V}_1))_{\{u_1, \ldots, u_r\}} \Rightarrow \{v_1, \ldots, v_r\} \sim \{u_1, \ldots, u_r\}$. Since $\rho_\lambda$ and $\phi_\lambda$ are injective, we have

$$R(E(\boldsymbol{A}, \boldsymbol{V}_1))_{\{v_1, \ldots, v_r\}} = R(E(\boldsymbol{A}, \boldsymbol{V}_1))_{\{u_1, \ldots, u_r\}}$$

$$\Rightarrow \forall \lambda \in [L] \quad \{S_{t, \{v_1, \ldots, v_r\}}\}_{\forall t \in [T] \text{ s.t. } \Lambda_t = \lambda} = \{S_{t, \{u_1, \ldots, u_r\}}\}_{\forall t \in [T] \text{ s.t. } \Lambda_t = \lambda}$$

$$\Rightarrow \forall j \in [T], \exists k \in [T] \quad \text{such that} \quad \{\boldsymbol{V}_{j, v_1}, \ldots, \boldsymbol{V}_{j, v_r}\} = \{\boldsymbol{V}_{k, u_1}, \ldots, \boldsymbol{V}_{k, u_r}\}$$

which means that $\forall j \in [T], \exists k \in [T]$ such that

$$\{f_\lambda(\boldsymbol{A}, \boldsymbol{V}_1 \oplus \mathbb{1}_{s_j})_{v_1}, \ldots, f_\lambda(\boldsymbol{A}, \boldsymbol{V}_1 \oplus \mathbb{1}_{s_j})_{v_r}\} = \{f_\lambda(\boldsymbol{A}, \boldsymbol{V}_1 \oplus \mathbb{1}_{s_k})_{u_1}, \ldots, f_\lambda(\boldsymbol{A}, \boldsymbol{V}_1 \oplus \mathbb{1}_{s_k})_{u_r}\}$$

where $\mathbb{1}_{s_j}$ (and similarly $\mathbb{1}_{s_k}$) is the vector with exactly $r-1$ ones (and $n - (r-1)$ zeros), at the $r-1$ positions specified by $s_j = \{j_1, \ldots, j_r\}$, with $j_1, \ldots, j_r$ node ids.

Since the above means that $\forall j \in [T], \exists k \in [T]$ where the sets are the same, then without loss of generality we have that

$$f_\lambda(\boldsymbol{A}, \boldsymbol{V}_1 \oplus \mathbb{1}_{s_j})_{v_1} = f_\lambda(\boldsymbol{A}, \boldsymbol{V}_1 \oplus \mathbb{1}_{s_k})_{u_1}$$

$$\vdots$$

$$f_\lambda(\boldsymbol{A}, \boldsymbol{V}_1 \oplus \mathbb{1}_{s_j})_{v_r} = f_\lambda(\boldsymbol{A}, \boldsymbol{V}_1 \oplus \mathbb{1}_{s_k})_{u_r}$$

Consider the first equation. Since $f_\lambda$ is node-most-expressive, then $v_1 \sim u_1$, which implies $\exists \boldsymbol{P}$ permuting nodes in $s_j$ with nodes in $s_k$ and $v_1$ with $u_1$ such that $\boldsymbol{A} = \boldsymbol{P} \boldsymbol{A} \boldsymbol{P}^T$ and $\boldsymbol{V}_1 \oplus \mathbb{1}_{s_k} = \boldsymbol{P}(\boldsymbol{V}_1 \oplus \mathbb{1}_{s_j})$. This implies that $(s_j, v_1) \sim (s_k, u_1)$ or, equivalently, $(j_1, \ldots, j_r, v_1) \sim (k_1, \ldots, k_r, u_1)$. Similarly, for all equations until the $r$-th equation we have that $(j_1, \ldots, j_r, v_r) \sim (k_1, \ldots, k_r, u_r)$. Consequently, it must be $(v_1, \ldots, v_r) \sim (u_1, \ldots, u_r)$.

($\Leftarrow$) We will show that $\{v_1, \ldots, v_r\} \sim \{u_1, \ldots, u_r\} \Rightarrow R(E(\boldsymbol{A}, \boldsymbol{V}_1))_{\{v_1, \ldots, v_r\}} = R(E(\boldsymbol{A}, \boldsymbol{V}_1))_{\{u_1, \ldots, u_r\}}$. Suppose by contradiction that $R(E(\boldsymbol{A}, \boldsymbol{V}_1))_{\{v_1, \ldots, v_r\}} \neq R(E(\boldsymbol{A}, \boldsymbol{V}_1))_{\{u_1, \ldots, u_r\}}$ which means that there exist $\lambda \in [L]$ such that $\boldsymbol{U}_{\lambda, \{v_1, \ldots, v_2\}} \neq \boldsymbol{U}_{\lambda, \{u_1, \ldots, u_r\}}$. Since $\phi_\lambda, \rho_\lambda$ are injective, without loss of generality this implies that $\exists j \in [T]$ such that

$$f_\lambda(\boldsymbol{A}, \boldsymbol{V}_1 \oplus \mathbb{1}_{s_j})_{v_1} \neq f_\lambda(\boldsymbol{A}, \boldsymbol{V}_1 \oplus \mathbb{1}_{s_k})_{u_1} \quad \forall k \in [n+1] \quad \text{such that} \quad \Lambda_j = \Lambda_k = \lambda$$

However, since $\{v_1, \ldots, v_r\} \sim \{u_1, \ldots, u_r\}$, $\exists k \in [T]$ with $\Lambda_k = \lambda$ such that considering $\boldsymbol{P}$ permuting $s_j$ with $s_k$ and $v_1$ with $u_1$ is such that $\boldsymbol{A} = \boldsymbol{P} \boldsymbol{A} \boldsymbol{P}^T$ and $\boldsymbol{V}_1 \oplus \mathbb{1}_{s_k} = \boldsymbol{P}(\boldsymbol{V}_1 \oplus \mathbb{1}_{s_j})$. Since $f_\lambda$ is structural, then

$$f_\lambda(\boldsymbol{A}, \boldsymbol{V}_1 \oplus \mathbb{1}_{s_j})_{v_1} = f_\lambda(\boldsymbol{A}, \boldsymbol{V}_1 \oplus \mathbb{1}_{s_k})_{u_1}$$

reaching a contradiction. $\square$

### B.4  AUXILIARY THEOREMS AND PROOFS

**Lemma B.5.** *Consider the specialization of our sequential breaking algorithm to the Lanczos algorithm* (c.f., Appendix B.2.1). *For every step $t$ such that $\beta_t = 0$, if $v_t$ is such that it exists another node $u$ with $\boldsymbol{V}_{i,v_t} = \boldsymbol{V}_{i,u}$ for all $i < t$, then $\boldsymbol{V}_t = \frac{\mathbb{1}_{v_t} - proj_{\boldsymbol{V}_1, \dots \boldsymbol{V}_{t-1}}(\mathbb{1}_{v_t})}{\|\mathbb{1}_{v_t} - proj_{\boldsymbol{V}_1, \dots \boldsymbol{V}_{t-1}}(\mathbb{1}_{v_t})\|} \in \mathbb{R}^n$ is a unit vector orthogonal to previous vectors.*

*Proof.* First, we note that for every step $t$, if $\beta_t = 0$, then[7]

$$\mathbf{w}_{t-1} = \mathbf{w}'_{t-1} - \alpha_{t-1}\boldsymbol{V}_{t-1} - \beta_{t-1}\boldsymbol{V}_{t-2} = \mathbf{0},$$

which implies

$$\mathbf{w}'_{t-1} = \boldsymbol{A}\boldsymbol{V}_{t-1} \in \text{span}(\boldsymbol{V}_{t-1}, \boldsymbol{V}_{t-2}).$$

Now we only need to show that $\mathbb{1}_{v_t} \notin \text{span}(\boldsymbol{V}_{t-1}, \dots, \boldsymbol{V}_1)$. Assume by contradiction that instead $\mathbb{1}_{v_t} \in \text{span}(\boldsymbol{V}_{t-1}, \dots, \boldsymbol{V}_1)$, and therefore it can be obtained as a linear combination of $\boldsymbol{V}_{t-1}, \dots, \boldsymbol{V}_1$. However, since by assumption $\exists u$ such that $\boldsymbol{V}_{i,v} = \boldsymbol{V}_{i,u}$ for all $i < t$, then the linear combination of $\boldsymbol{V}_{t-1}, \dots, \boldsymbol{V}_1$ will necessarily result in a vector having the same entry for $v_t$ and $u$, and therefore cannot be $\mathbb{1}_{v_t}$ which is 1 in $v_t$ and 0 in $u$, leading to our contradiction. □

**Theorem B.6.** *Consider the specialization of our sequential breaking algorithm to the Lanczos algorithm* (c.f., Appendix B.2.1). *There exist most-expressive breaking selectors $(h_\lambda^\star)_{\lambda=1}^L$ (c.f., Definition A.1) which are sufficient to always obtain breaking nodes $v_t$ such that, when $\beta_t = 0$, then $\boldsymbol{V}_t = \frac{\mathbb{1}_{v_t} - proj_{\boldsymbol{V}_1, \dots \boldsymbol{V}_{t-1}}(\mathbb{1}_{v_t})}{\|\mathbb{1}_{v_t} - proj_{\boldsymbol{V}_1, \dots \boldsymbol{V}_{t-1}}(\mathbb{1}_{v_t})\|} \in \mathbb{R}^n$ is a unit vector orthogonal to previous vectors.*

*Proof.* The proof follows directly from Lemma B.5 by showing that there exist most-expressive $(h_\lambda^\star)_{\lambda=1}^L$ satisfying its assumptions. These breaking selectors can be easily constructed in the following way:

(1) For the first $t$ such that $\beta_t = 0$, we need the corresponding $h_\lambda^\star$ to return an isomorphism class that contains more than one node. In other words, the length of the list must be greater than 1 ($k_\lambda^\star > 1$). Since this is the first time we break symmetries, this ensure that there exists another $u$ satisfying Lemma B.5.

(2) For any other $t$ such that $\beta_t = 0$, we also need $h_\lambda^\star$ to be returning a non-singleton isomorphism class. This class must be such that it exists another node $u$ isomorphic to $v_t$ such that $\boldsymbol{V}_{i,v_t} = \boldsymbol{V}_{i,u}$ for all $i < t$, so that the assumptions of Lemma B.5 are met. Note that there always exist such $h_\lambda^\star$, as we show next.

Assume by contradiction that there isn't any $v_t$ such that $\exists u$ with $\boldsymbol{V}_{i,v_t} = \boldsymbol{V}_{i,u}$ for all $i < t$. Then it is always possible to obtain a new orthogonal vector using the Lanczos algorithm (by propagating the unique features), which implies that $\beta_t \neq 0$ reaching a contradiction.

Finally, note that such $(h_\lambda^\star)_{\lambda=1}^L$ are non overlapping: whenever an $h_\lambda^\star$ is chosen to obtain a corresponding $v_t$, then the symmetry is broken and therefore using again the isomorphism class as output of another $h_{\lambda'}^\star$, with $\lambda' > \lambda$, will not result in additional symmetry breakings, implying we need to use a different non-singleton isomorphism class for later $t' > t$ such that $\beta_{t'} = 0$.

□

**Theorem B.7.** *Let $G_1 = (\boldsymbol{A}_1, \boldsymbol{X}_1)$, $G_2 = (\boldsymbol{A}_2, \boldsymbol{X}_2)$ be two isomorphic graphs, i.e., $\boldsymbol{A}_2 = \boldsymbol{P}\boldsymbol{A}_1\boldsymbol{P}^T$, $\boldsymbol{X}_2 = \boldsymbol{P}\boldsymbol{X}_1$, for some permutation matrix $\boldsymbol{P}$. Denote the Lanczos vectors of $G_1$ as $\boldsymbol{V}_1^{G_1}, \dots, \boldsymbol{V}_n^{G_1}$, and the Lanczos vectors of $G_2$ as $\boldsymbol{V}_1^{G_2}, \dots, \boldsymbol{V}_n^{G_2}$, obtained by our sequential breaking algorithm with $\boldsymbol{V}_1^{G_1}$ and $\boldsymbol{V}_1^{G_2}$ the normalized outputs of the structural function $g_{struc}$ (Equation (1)) and $(h_\lambda)_{\lambda=1}^L$ the most-expressive breaking selectors (Definition A.1). There exists a permutation matrix $\boldsymbol{B}$ such that*

$$\boldsymbol{V}_t^{G_2} = \boldsymbol{B}\boldsymbol{V}_t^{G_1}, \qquad \forall t \leq n. \tag{16}$$

---

[7]Note that if $t = 2$, then $\boldsymbol{w}_1 = \boldsymbol{w}'_1 - \alpha_1\boldsymbol{V}_1$ and the equation is valid by assuming $\boldsymbol{V}_{t-2} = \mathbf{0}$, $\beta_1 = 0$.

*Proof.* For simplicity of notation, let $b_1 = \text{Concat}((h_\lambda(A_1, V_1))_{\lambda=1}^L)$ and $b_2 = \text{Concat}((h_\lambda(A_2, V_2))_{\lambda=1}^L)$ be the two lists of breakings. Since $b_1, b_2$ are simply orderings of node ids, they can equivalently be represented by permutation matrices, which we will denote by $P_{b_1}$ and $P_{b_2}$. Let $B$ the permutation matrix transforming $b_1$ into $b_2$, that is:

$$B = P_{b_2}^T P_{b_1}. \tag{17}$$

We will prove the result by induction by showing that Equation (16) is true with $B$ as in Equation (17).

**Base case.** Consider $t = 1$. Since $(h_\lambda)_{\lambda=1}^L$ are most-expressive and structural, the breaking nodes in the same position in $b_1$ and $b_2$ are isomorphic. Therefore, $B$ maps each node in $G_1$ to a node in $G_2$ isomorphic to it, which implies

$$BV_1^{G_1} = V_1^{G_2}.$$

Consider $t = 2$. We distinguish two cases: (1) $\beta_2^{G_1} \neq 0$ and $\beta_2^{G_2} \neq 0$ or (2) $\beta_2^{G_1} = 0$ and $\beta_2^{G_2} = 0$. Indeed, it is not possible to have $\beta_2^{G_1} \neq \beta_2^{G_2}$, as they are obtained from $V_1^{G_1}$ and $V_1^{G_2}$ using a structural function.

If $\beta_2 \neq 0$, since the steps of the Lanczos algorithm outside the else-statement are all structural, since $V_1^{G_2}$ is simply a permutation of $V_1^{G_1}$, and since $B$ maps each node in $G_1$ to a node in $G_2$ isomorphic to it, then again

$$BV_2^{G_1} = V_2^{G_2}.$$

If, instead, $\beta_2^{G_1} = \beta_2^{G_2} = 0$, then, for each graph $G_i$, $i \in \{1, 2\}$, given a breaking node $v_2^{G_i}$ we first construct the breaking vector as $U_2^{G_i} = \mathbb{1}_{v_2^{G_i}}$ and then orthonormalize it using Gram-Schmidt. Note that since $U_2^{G_1}, U_2^{G_2}$ are standard basis vectors, which differ in that they are obtained by potentially different breaking nodes $v_2^{G_1}$ and $v_2^{G_2}$, we have $BU_2^{G_1} = U_2^{G_2}$ by definition of $B$ (which indeed maps $v_2^{G_1}$ into $v_2^{G_2}$). Therefore, denoting by $\hat{V}_2^{G_i}$ the unnormalized version of $V_2^{G_i}$, we have

$$
\begin{aligned}
B\hat{V}_2^{G_1} &= B(U_2^{G_1} - \text{proj}_{V_1^{G_1}}(U_2^{G_1})) \\
&= BU_2^{G_1} - B\text{proj}_{V_1^{G_1}}(U_2^{G_1}) \\
&= BU_2^{G_1} - B(U_2^{G_1 T} V_1^{G_1}) V_1^{G_1} \\
&= BU_2^{G_1} - (U_2^{G_1 T} B^T B V_1^{G_1}) B V_1^{G_1} \\
&= BU_2^{G_1} - \text{proj}_{BV_1^{G_1}}(BU_2^{G_1}) \\
&= U_2^{G_2} - \text{proj}_{V_1^{G_2}}(U_2^{G_2}) \\
&= \hat{V}_2^{G_2}.
\end{aligned}
$$

Since we have just shown that Equation (16) is true for the unnormalized vectors, then dividing both sides by the norm preserves the relationship, and therefore

$$BV_2^{G_1} = V_2^{G_2}.$$

**Inductive step.** Consider a step $t - 1 < n$, and suppose

$$V_i^{G_2} = BV_i^{G_1}, \qquad \forall i \leq t - 1. \tag{18}$$

We next show that $V_t^{G_2} = BV_t^{G_1}$. We distinguish again two cases, $\beta_t \neq 0$ and $\beta_t = 0$.

If $\beta_t \neq 0$, then the step to obtain the next Lanczos vector is a structural function, and since $B$ maps each node in $G_1$ to a node in $G_2$ isomorphic to it, then again

$$BV_t^{G_1} = V_t^{G_2}.$$

If $\beta_t = 0$, then $U_t^{G_1}, U_t^{G_2}$ are standard basis vectors, which differ in that they are obtained by potentially different breaking nodes $v_t^{G_1}$ and $v_t^{G_2}$, and again we have $BU_t^{G_1} = U_t^{G_2}$ by definition of $B$. Therefore, denoting by $\hat{V}_t^{G_i}$ the unnormalized version of $V_t^{G_i}$,

$$B\hat{V}_t^{G_1} = B(U_t^{G_1} - \sum_{i=1}^t \text{proj}_{V_i^{G_1}}(U_t^{G_1}))$$

$$= BU_t^{G_1} - B \sum_{i=1}^{t} \text{proj}_{V_i^{G_1}}(U_t^{G_1})$$

$$= BU_t^{G_1} - B \sum_{i=1}^{t} (U_t^{G_1 T} V_i^{G_1}) V_i^{G_1}$$

$$= BU_t^{G_1} - \sum_{i=1}^{t} (U_t^{G_1 T} B^T B V_i^{G_1}) B V_i^{G_1}$$

$$= BU_t^{G_1} - \sum_{i=1}^{t} \text{proj}_{BV_i^{G_1}}(BU_t^{G_1})$$

$$= U_t^{G_2} - \sum_{i=1}^{t} \text{proj}_{V_i^{G_2}}(U_t^{G_2})$$

$$= \hat{V}_t^{G_2}$$

which implies that

$$BV_t^{G_1} = V_t^{G_2},$$

since we are simply dividing both sides by the norm. $\qquad\square$

*Remark* B.8 (Equivariance of $V_T, \ldots, V_1$). Theorem B.7 demonstrates that the vectors $V_T, \ldots, V_1$ produced by the expansion map permute with the input graph. In other words, for any two isomorphic graphs $G_1$, $G_2$, there exists a permutation matrix $B$ such that $V_t^{G_2} = BV_t^{G_1}$, for all $t \leq T$. This property holds for both the Lanczos vectors and any other structural function $f_t$. In the latter case, the proof follows from Theorem B.7 by simply disregarding the $\beta = 0$ case.

## C  RELATED WORK

**Towards Graph Foundation Models.**   Unlike vision and language, graph machine learning does not yet have a standard recipe for training general purpose models. Because of this, there is an active debate on what ingredients are needed (Mao et al., 2024). Some requirements are clear, such as the need for a single input feature space common to all graphs, akin to a language model tokenizer (Radford et al., 2019). Initial approaches include "textifying" all node features and using pre-trained text encoders to produce feature vectors (Chen et al., 2024b; Liu et al., 2024), while recent works have proposed different solutions for addressing the problem of differing features across graphs (Zhao et al., 2023; Lachi et al., 2024; Zhao et al., 2024; Xia & Huang, 2024; He & Hooi, 2024; Shen et al., 2024; Hou et al., 2024), including for cases where the differences are only the relation identifiers (Gao et al., 2023; Zhou et al., 2023; Galkin et al., 2024). Our work focuses on another crucial need: developing general-purpose representations capable of performing multiple tasks. To best of our knowledge, there is no existing GNN-based model capable of learning representations that are suitable for different tasks. Existing effort are either domain-specific (Zheng et al., 2024; Beaini et al., 2024; Sypetkowski et al., 2024), or heavily rely on LLMs (Huang et al., 2024a; Liu et al., 2024; Chen et al., 2024a; Tang et al., 2024), which can present efficiency challenges (Mao et al., 2024). Perhaps more crucially, no existing method offers guarantees on the expressiveness of the learned representations. Our work builds on this growing understanding of GNN expressive power (Morris et al., 2023), moving from expressivity for a single task to expressivity across many tasks.

**Associative Memories and Distributed Representations.**   Earlier neural network blueprints broadly fell into one of two extreme camps: "one concept one neuron", or distributed (*holographic*) representations, where an individual concept is spread across many neurons. Arguments for either side were advanced partially through appeal neuroscience's understanding the animal brains. The deep learning era triggered by AlexNet (Krizhevsky et al., 2012) largely set this debate to one side, as modern neural networks tended to exhibit properties somewhere in the middle, with neuron-level concepts recoverable in startling detail in some cases (Templeton et al., 2024) and superimposed in other cases (Elhage et al., 2022). Our effort to explicitly reintroduce holographic representations for graphs is in response to the different structural properties (namely symmetries) of graph data compared to vision, language and other modalities.

**Symmetry Breaking Algorithms.** The expansion map in HoloGNN uses symmetry breakings as a key subroutine to produce general and expressive representations. Symmetry breakings is often used in GNNs that "color" or "mark" individual nodes in order to distinguish them, a common form of positional encoding (You et al., 2019; Cui et al., 2022), oftentimes also obtained by sampling random features for each node (You et al., 2019; Murphy et al., 2019; Abboud et al., 2021; Sato et al., 2021; Puny et al., 2020). Symmetry breakings has also been used as a form of distance encoding, which can be specialized for predictions of different tasks (Li et al., 2020; Yin et al., 2020). In contrast, our approach uses the same symmetry breaking for all tasks, with task specialization occurring only in the (lightway) reduction map. But symmetry breaking has been used more broadly in the analysis of graphs. Examples include in algebraic combinatorics to study the automorphism groups of different graph classes (Albertson & Collins, 1996), and in combinatorial optimization, where symmetry breaking is used to simplify complex combinatorial constraint sets (Fahle et al., 2001). Moreover, symmetry breaking is also used in gold-standard linear algebra algorithms, such as Lanczos and Arnoldi iterations for computing eigendecompositions (Lanczos, 1950; Arnoldi, 1951). In this case, symmetry breaking is used to (randomly) resolve ambiguities in eigenspaces for eigenvalues with repeated roots. Stepping back, although symmetry breaking is a known concept in graph learning, our approach distinguishes itself by ensuring that symmetry breaking is performed in such a way that it is possible to re-introduce the lost permutation symmetries and produce structural representations.

**Subgraph GNNs.** The concept of symmetry breaking, particularly through our parallel breaking algorithm, shares similarities with the recent works on Subgraph GNNs (Zhang & Li, 2021; Cotta et al., 2021; Papp et al., 2021; Bevilacqua et al., 2022; Zhao et al., 2022; Papp & Wattenhofer, 2022; Frasca et al., 2022; Qian et al., 2022; Zhang et al., 2023; Bevilacqua et al., 2024; Bar-Shalom et al., 2024a;b), which also break symmetries by marking nodes to enhance the expressive power of GNNs, as originally proposed in Li et al. (2020). However, our approach significantly differs from Subgraph GNNs in that our aim is not to produce expressive representations for a single task, but rather to pre-train models returning representations that can be used for any task. To this end, we introduce the concept of a reduction map, which generates structural representations suitable for any task order. In contrast, Subgraph GNNs use their representations to revert to structural node or graph-level representation. Additionally, while Subgraph GNNs generally break symmetries for all nodes in the graph, our expansion map can leverage $T^\star \ll n$, significantly improving efficiency.

**Graph Transformers.** Finally, we acknowledge the growing interest in graph transformer models (Kreuzer et al., 2021; Dwivedi & Bresson, 2021; Ying et al., 2021; Rampášek et al., 2022; Kim et al., 2022; Müller et al., 2024), driven by their strong performance especially on graph classification tasks. However, graph transformers rely on positional encodings (Dwivedi et al., 2023), which make their learned representations inherently positional. As discussed in Section 2, positional representations are task-order specific. This task-order specificity makes graph transformers unsuitable for pre-training a single model that can generate flexible representations capable of adapting to tasks of varying orders. In contrast, HoloGNN focuses on producing task-agnostic, symmetry-free representations that can efficiently adapt to different task orders.

## D  COMPLEXITY ANALYSIS

In this section, we analyze the complexity of the symmetry-breaking algorithms (Algorithm 1 and Algorithm 2), which represent the foundation of our HoloGNN.

The complexity of the algorithms is determined by the complexity of $f_t$ (for sequential breaking) and $f_\lambda$ (for parallel breaking). Suppose $f_t$ and $f_\lambda$ are Message Passing Neural Networks (MPNNs), which have linear complexity in the number of nodes $n$ and edges $m$, $\mathcal{O}(n + m)$, where the feature dimension $d$ is considered constant. Since both sequential and parallel breaking perform $T$ calls to the corresponding MPNNs, each time by passing a new breaking node, the complexity is $\mathcal{O}(T(n + m))$ in both cases. Note that in our experiments $T \ll n$ and $T \ll m$. For instance, in the Pubmed dataset, $T = 8$, $n = 19717$, $m = 88648$. Finally, we note that in the parallel breaking algorithm, the additional $T$ factor in the complexity can effectively be mitigated, as all $T$ calls can be done in parallel. This is because each breaking is applied to $\mathbf{V}_1$, and $f_\lambda$ does not take all prior embeddings.

Table 3: Performance on the MovieLens dataset in each task (column), when pre-trained in each of the four tasks (row block). The diagonal (light blue) contains the performance of the models when trained and tested on the same task. "OOM" indicates out of RAM, and "NA" indicates that either the task could not be performed due to out-of-memory issues or because the model cannot perform that task order. *Each column illustrates the performance drop when comparing the results of pre-training on the target task with pre-training on all other tasks, with HoloGNN variants exhibiting significantly smaller drops.*

| | | MOVIELENS | | | |
|---|---|---|---|---|---|
| | | **UserMovie** (RMSE ↓) | **MovieMovie** (Acc. ↑) | **UserMovieUser** (Acc. ↑) | **UserMovieMovie** (Acc. ↑) |
| **UserMovie** | SEAL | $0.960 \pm 0.010$ | $0.815 \pm 0.002$ | OOM | OOM |
| | NBFNet (no feat) | $1.044 \pm 0.004$ | $0.981 \pm 0.001$ | NA | NA |
| | NBFNet | $0.967 \pm 0.005$ | $0.892 \pm 0.001$ | NA | NA |
| | SAGE | $0.943 \pm 0.006$ | $0.657 \pm 0.008$ | $0.595 \pm 0.002$ | $0.589 \pm 0.001$ |
| | SAGE & LapPE | $0.949 \pm 0.019$ | $0.765 \pm 0.004$ | $0.593 \pm 0.002$ | $0.592 \pm 0.001$ |
| | HoloGNN (MP) | $0.949 \pm 0.002$ | $0.921 \pm 0.001$ | $0.708 \pm 0.001$ | $0.808 \pm 0.002$ |
| | HoloGNN (GNN) | $0.943 \pm 0.001$ | $0.918 \pm 0.002$ | $0.741 \pm 0.003$ | $0.810 \pm 0.001$ |
| **MovieMovie** | SEAL | $1.044 \pm 0.007$ | $0.981 \pm 0.001$ | OOM | OOM |
| | NBFNet (no feat) | $1.053 \pm 0.007$ | $0.989 \pm 0.001$ | NA | NA |
| | NBFNet | $1.050 \pm 0.007$ | $0.989 \pm 0.001$ | NA | NA |
| | SAGE | $1.052 \pm 0.004$ | $0.977 \pm 0.002$ | $0.635 \pm 0.006$ | $0.667 \pm 0.002$ |
| | SAGE & LapPE | $1.045 \pm 0.003$ | $0.975 \pm 0.003$ | $0.646 \pm 0.016$ | $0.675 \pm 0.001$ |
| | HoloGNN (MP) | $1.019 \pm 0.005$ | $0.984 \pm 0.001$ | $0.694 \pm 0.001$ | $0.810 \pm 0.001$ |
| | HoloGNN (GNN) | $1.020 \pm 0.006$ | $0.986 \pm 0.001$ | $0.721 \pm 0.001$ | $0.810 \pm 0.004$ |
| **UserMovieUser** | SEAL | NA | NA | OOM | NA |
| | NBFNet (no feat) | NA | NA | NA | NA |
| | NBFNet | NA | NA | NA | NA |
| | SAGE | $1.014 \pm 0.005$ | $0.733 \pm 0.002$ | $0.816 \pm 0.003$ | $0.646 \pm 0.001$ |
| | SAGE & LapPE | $1.012 \pm 0.008$ | $0.793 \pm 0.002$ | $0.812 \pm 0.006$ | $0.647 \pm 0.001$ |
| | HoloGNN (MP) | $1.005 \pm 0.002$ | $0.936 \pm 0.003$ | $0.794 \pm 0.004$ | $0.779 \pm 0.001$ |
| | HoloGNN (GNN) | $0.991 \pm 0.005$ | $0.972 \pm 0.001$ | $0.811 \pm 0.002$ | $0.795 \pm 0.001$ |
| **UserMovieMovie** | SEAL | NA | NA | NA | OOM |
| | NBFNet (no feat) | NA | NA | NA | NA |
| | NBFNet | NA | NA | NA | NA |
| | SAGE | $1.007 \pm 0.014$ | $0.930 \pm 0.003$ | $0.721 \pm 0.001$ | $0.890 \pm 0.007$ |
| | SAGE & LapPE | $0.995 \pm 0.004$ | $0.900 \pm 0.002$ | $0.719 \pm 0.001$ | $0.885 \pm 0.008$ |
| | HoloGNN (MP) | $0.982 \pm 0.005$ | $0.974 \pm 0.002$ | $0.756 \pm 0.002$ | $0.904 \pm 0.009$ |
| | HoloGNN (GNN) | $0.989 \pm 0.004$ | $0.974 \pm 0.002$ | $0.769 \pm 0.001$ | $0.905 \pm 0.008$ |

(Row blocks labeled vertically under PRE-TRAINING TASK: UserMovie, MovieMovie, UserMovieUser, UserMovieMovie)

# E ADDITIONAL EXPERIMENTS AND DETAILS

## E.1 ADDITIONAL EXPERIMENTS

In this section, we provide additional insights and results on the experiments. We also introduce an additional baseline, created by concatenating structural and positional encodings—referred to as SAGE & LapPE for MovieLens and GCN & LapPE for Planetoid, using SAGE or GCN for structural encodings and Laplacian eigenvectors for positional encodings (Dwivedi et al., 2023). This baseline demonstrates that simple concatenation does not achieve the performance of HoloGNN.

### E.1.1 MOVIELENS

Table 3 compares the performance of HoloGNN in the MovieLens dataset. Each column represents a different task we test on, namely **UserMovie**, **MovieMovie**, **UserMovieUser**, **UserMovieMovie**. Each row block corresponds to a pre-training task, with each row being one model. This implies that each entry in the table corresponds to the performance of a model (row) when pre-trained on one task (row block) and adapted to a potentially different task (column). The diagonals, colored in light blue, contain the performance of each model when pre-trained and tested *on the same task*; in other words, each diagonal shows the performance of the supervised version of each model.

We consider four tasks, **UserMovie**, **MovieMovie** which are order-2 tasks, and **UserMovieUser**, **UserMovieMovie**, which are order-3 tasks. In particular, the **UserMovie** task, the standard task for this dataset, predicts a user's rating of a movie, **MovieMovie** predicts the existence of a metapath between two movies, **UserMovieUser** predicts if two users will both watch a particular movie,

Table 4: Performance on the Planetoid datasets in each of the two tasks (column), when pre-trained in either of the two tasks (row block). For each dataset, the diagonal (light blue) shows the performance of the models when trained and tested on the same task. *Each column illustrates the performance drop when comparing the results of pre-training on the target task with pre-training on the other task. HoloGNN variants demonstrate more consistent performance, with significantly smaller drops than all other methods especially when pre-trained on link prediction and tested on node classification.*

| | | | CORA | | CITESEER | | PUBMED | |
|---|---|---|---|---|---|---|---|---|
| | | | Node Class. (Acc. ↑) | Link Pred. (AUROC ↑) | Node Class. (Acc. ↑) | Link Pred. (AUROC ↑) | Node Class. (Acc. ↑) | Link Pred. (AUROC ↑) |
| PRE-TRAINING TASK | Node Class. | SEAL | $77.2_{\pm 2.1}$ | $85.9_{\pm 1.6}$ | $64.5_{\pm 0.5}$ | $88.8_{\pm 1.1}$ | $75.0_{\pm 1.1}$ | $93.3_{\pm 0.2}$ |
| | | NBFNet (no feat) | $31.3_{\pm 1.3}$ | $82.1_{\pm 0.9}$ | $25.7_{\pm 1.6}$ | $89.6_{\pm 1.1}$ | $40.5_{\pm 0.5}$ | $92.5_{\pm 0.3}$ |
| | | NBFNet | $68.8_{\pm 1.3}$ | $87.5_{\pm 1.6}$ | $50.4_{\pm 2.8}$ | $88.8_{\pm 1.2}$ | $71.4_{\pm 2.1}$ | $95.7_{\pm 0.3}$ |
| | | GCN | $81.4_{\pm 1.0}$ | $89.9_{\pm 0.2}$ | $70.1_{\pm 0.7}$ | $89.5_{\pm 0.8}$ | $79.0_{\pm 0.2}$ | $93.2_{\pm 0.1}$ |
| | | GCN & LapPE | $81.5_{\pm 0.8}$ | $90.2_{\pm 0.8}$ | $70.0_{\pm 0.4}$ | $88.5_{\pm 0.8}$ | $78.9_{\pm 0.3}$ | $93.5_{\pm 0.1}$ |
| | | HoloGNN (MP) | $81.5_{\pm 0.8}$ | $91.0_{\pm 0.3}$ | $70.0_{\pm 0.2}$ | $93.7_{\pm 0.1}$ | $78.7_{\pm 0.2}$ | $93.5_{\pm 0.2}$ |
| | | HoloGNN (GNN) | $81.6_{\pm 0.7}$ | $91.1_{\pm 0.4}$ | $70.1_{\pm 0.8}$ | $94.0_{\pm 0.6}$ | $78.7_{\pm 0.8}$ | $93.8_{\pm 0.1}$ |
| | Link Pred. | SEAL | $35.6_{\pm 2.6}$ | $93.5_{\pm 0.5}$ | $31.6_{\pm 0.8}$ | $90.6_{\pm 1.1}$ | $44.4_{\pm 1.1}$ | $97.8_{\pm 0.2}$ |
| | | NBFNet (no feat) | $26.7_{\pm 3.3}$ | $94.8_{\pm 0.7}$ | $22.8_{\pm 2.8}$ | $92.4_{\pm 1.6}$ | $40.5_{\pm 0.3}$ | $98.3_{\pm 0.2}$ |
| | | NBFNet | $27.8_{\pm 6.1}$ | $89.0_{\pm 2.7}$ | $25.0_{\pm 1.0}$ | $85.2_{\pm 1.2}$ | $56.7_{\pm 0.2}$ | $98.2_{\pm 0.2}$ |
| | | GCN | $74.2_{\pm 0.2}$ | $89.6_{\pm 0.5}$ | $68.3_{\pm 0.2}$ | $91.4_{\pm 0.2}$ | $76.8_{\pm 0.1}$ | $96.0_{\pm 0.7}$ |
| | | GCN & LapPE | $73.4_{\pm 0.6}$ | $89.3_{\pm 1.8}$ | $68.1_{\pm 0.5}$ | $91.1_{\pm 0.7}$ | $75.3_{\pm 0.6}$ | $95.8_{\pm 0.4}$ |
| | | HoloGNN (MP) | $80.5_{\pm 0.8}$ | $92.5_{\pm 0.5}$ | $69.0_{\pm 0.2}$ | $94.9_{\pm 0.5}$ | $79.7_{\pm 0.4}$ | $96.7_{\pm 0.4}$ |
| | | HoloGNN (GNN) | $80.2_{\pm 0.5}$ | $93.9_{\pm 0.1}$ | $69.0_{\pm 0.9}$ | $95.0_{\pm 0.5}$ | $79.8_{\pm 0.5}$ | $97.2_{\pm 0.8}$ |

**UserMovieMovie**, if a user will watch two particular movies. For SEAL, we use OOM to denote that the model went out of RAM memory in the preprocessing phase, when constructing the subgraphs. We then use NA in the other tasks as the embeddings cannot be adapted to any other tasks, since they could not be learned. For NBFNet and NFBNet (no feat) (the variant that does not use node features) we use NA in the order-3 tasks as NBFNet cannot handle tasks of order greater than 2.

As can be seen from Table 3, HoloGNN (GNN) and its variant HoloGNN (MP), which uses simple message passing propagations as $f_\lambda$ instead of a GNN network, exhibit a significantly smaller gap in the performance when pre-trained on the task and when adapted from a different one. For example, when considering the order-3 task **UserMovieUser**, the difference in performance between HoloGNN (GNN) when pre-trained on the task ($0.811$) and its worst performance when pre-trained in any other task ($0.721$, obtained pre-training on the order-2 **MovieMovie**) results in a performance drop of around $9\%$. SAGE shows a much larger drop of around $22\%$, with its accuracy dropping from $0.816$ when pre-trained on **UserMovieUser** to $0.595$ when pre-trained on the order-2 **User-Movie** (worst performance when pre-trained in any other task). For the order-2 task **MovieMovie**, HoloGNN exhibits a drop of $7\%$, while SAGE's drop becomes $32\%$. Note that this trend also holds true for all other methods, with SEAL and NBFNet consistently showing larger performance gaps than HoloGNN. The only exception is NBFNet (no feat) in the **MovieMovie** dataset when pre-trained on **UserMovie**; however, this is due to NBFNet underperforming on the pre-training task, suggesting it is merely capturing node distances rather than learning the task, which happens to suffice for **MovieMovie**.

### E.1.2 PLANETOID

Table 4 compares the performance of HoloGNN in the Planetoid datasets Cora, Citeseer and Pubmed. Each column block represents a different dataset, and each column corresponds to a different task we test on, namely **Node Classification** and **Link Prediction**. Each row block corresponds to a pre-training task, with each row being one of the models. This implies that each entry in the table corresponds to the performance of a model (row) when pre-trained on one task (row block) and adapted to a potentially different task (column), with *both tasks from the same dataset*. The diagonal for each dataset, which is colored in light blue, contains the performance of each model when pre-trained and tested *on the same task*; in other words, it contains to the performance of the supervised version of each model in that dataset.

For each dataset, we consider the order-1 task **Node Classification** and the order-2 task **Link Prediction**. As can be seen from Table 4, HoloGNN (GNN) and its variant HoloGNN (MP), which uses

simple message passing propagations as $f_\lambda$ instead than a shallow GNN network, demonstrate significantly more consistent performance compared to other methods. While all other models exhibit a significant performance loss when pre-trained on **Link Prediction** and adapted to **Node Classification**, this is not the case for HoloGNN and HoloGNN (MP). For instance, in the CORA dataset, the difference in performance between HoloGNN (GNN) when pre-trained on **Node Classification** (81.6%) and its performance when pre-trained on **Link Prediction** and adapted to **Node Classification** (80.2%) results in a performance drop of around 1%. On the contrary, SAGE suffers from a performance drop of 7%, while NBFNet and SEAL show a more dramatic drop of 41%. Pre-training on **Node Classification** results in less dramatic drops, especially for SAGE which obtains node representation (structural of order 1) both for node- and link-level tasks, which therefore tend to yield similar performances. Nonetheless, HoloGNN still demonstrate smaller performance drops, and overall better performances.

### E.1.3 MOLHIV

We conduct an additional experiment to evaluate the performance of HoloGNN when adapted to a different task order, namely the order-n task graph classification. Since there is currently no standard dataset that includes both graph classification and another type of task, we considered the molhiv dataset from the OGB benchmark (Hu et al., 2020), which is designed for graph classification (an order-n task) and we created a link prediction task (an order-2 task) on the dataset. We evaluated the performance of both GCN and HoloGNN when pre-trained on link prediction and adapted to graph classification and compare them to their performance when directly trained and evaluated on the graph classification task.

Table 5: Performance on molhiv on the graph classification task, when pre-trained in either graph classification or link prediction. *HoloGNN demonstrates more consistent performance, with significantly smaller drops than GCN.*

|  | MOLHIV **Graph Class.** (ROC-AUC ↑) |
|---|---|
| GCN (Pretrain on Graph Class) | $76.06_{\pm0.97}$ |
| HoloGNN (GNN) (Pretrain on Graph Class) | $78.19_{\pm0.91}$ |
| GCN (Pretrain on Link Pred) | $70.29_{\pm1.33}$ |
| HoloGNN (GNN) (Pretrain on Link Pred) | $76.43_{\pm1.20}$ |

Results are reported in Table 5. The key observation is that the difference in performance between HoloGNN trained on graph classification (78.19%) vs pre-trained on link prediction and adapted to graph classification (76.43%) results in a performance drop of only 1.76%, while GCN suffers a more pronounced drop of 5.77%. These results further showcase the ability of HoloGNN to pre-train on one task order and adapt effectively to other task orders.

### E.1.4 ABLATION STUDY ON THE NUMBER OF BREAKINGS

We additionally evaluate the impact of the number of breakings $T$. We consider the Cora dataset and report results of HoloGNN for different values of $T$ on the node classification task, both when trained on it and when pre-trained on link prediction and adapted to node classification. Table 6 shows that small $T$ values are sufficient for obtaining representations that can effectively adapt to different tasks, and increasing $T$ results in marginal improvements. Indeed, regardless of the value of $T$, the performance drop of HoloGNN when trained on node classification and when trained on link prediction and adapted to node classification is significantly smaller than the performance drop suffered by GCN.

Table 6: Impact of the number of breakings $T$ in HoloGNN. Regardless of the value of $T$, the performance drop of HoloGNN when trained on node classification and when trained on link prediction and adapted to node classification is significantly smaller than the performance drop suffered by GCN, with larger values of $T$ yielding marginal improvements.

|  | CORA **Node Class.** (Acc. ↑) |
|---|---|
| GCN (Pretrain on Node Class) | $81.4_{\pm1.0}$ |
| HoloGNN (GNN) $T = 4$ (Pretrain on Node Class) | $80.9_{\pm0.8}$ |
| HoloGNN (GNN) $T = 8$ (Pretrain on Node Class) | $81.6_{\pm0.7}$ |
| HoloGNN (GNN) $T = 64$ (Pretrain on Node Class) | $82.1_{\pm0.1}$ |
| GCN (Pretrain on Link Pred) | $74.2_{\pm0.2}$ |
| HoloGNN (GNN) $T = 4$ (Pretrain on Link Pred) | $79.4_{\pm0.7}$ |
| HoloGNN (GNN) $T = 8$ (Pretrain on Link Pred) | $80.2_{\pm0.5}$ |
| HoloGNN (GNN) $T = 64$ (Pretrain on Link Pred) | $80.5_{\pm0.7}$ |

Table 7: Comparison with GPS (Rampášek et al., 2022) on CORA in each of the two tasks (column), when pre-trained in either of the two tasks. *HoloGNN demonstrates more consistent performance, with significantly smaller drops than GPS when pre-trained on link prediction and adapted to node classification.*

| | CORA | |
| --- | --- | --- |
| | **Node Class.** (Acc. ↑) | **Link Pred.** (AUROC ↑) |
| GCN (Pretrain on Node Class) | $81.4 \pm 1.0$ | $89.9 \pm 0.2$ |
| GPS (Pretrain on Node Class) | $78.3 \pm 1.0$ | $93.0 \pm 0.1$ |
| HoloGNN (GNN) (Pretrain on Node Class) | $81.6 \pm 0.7$ | $91.1 \pm 0.4$ |
| GCN (Pretrain on Link Pred) | $74.2 \pm 0.2$ | $89.6 \pm 0.5$ |
| GPS (Pretrain on Link Pred) | $36.8 \pm 2.0$ | $92.7 \pm 0.4$ |
| HoloGNN (GNN) (Pretrain on Link Pred) | $80.2 \pm 0.5$ | $93.9 \pm 0.1$ |

### E.1.5 COMPARISON WITH GPS GRAPH TRANSFORMER

We additionally compare the performance of the GPS graph transformer (Rampášek et al., 2022) baseline with HoloGNN when adapted to a different task order than the one used for pre-training, using the CORA dataset.

Results are reported in Table 7, where each column represents a different task (Node Classification or Link Prediction). For each task, we evaluate the performance of the models when trained directly on that task (shown in light blue) and when pre-trained on the other task and adapted to the target task. While GPS performs well when adapted to Link Prediction, it suffers from a dramatic performance drop of $41.5\%$ when pre-trained on Link Prediction and adapted to Node Classification, while HoloGNN results in a performance drop of only around $1\%$.

### E.2 EXPERIMENTAL DETAILS

We implemented HoloGNN using Pytorch (Paszke et al., 2019) and Pytorch Geometric (Fey & Lenssen, 2019), and performed hyperparameter tuning using the Weight and Biases framework (Biewald, 2020). We ran our experiments on NVIDIA A100 and GeForce RTX 4090 GPUs.

In all our experiments except the eigenvector task, we implemented the parallel breaking algorithm. We use a feature-marking technique to break nodes, adding an additional dimension to the embeddings with value 1 if that node is selected to be broken, and 0 otherwise. Note that we break nodes one at a time, so there are $T$ distinct forward passes, as described in the parallel breaking algorithm (Algorithm 2). We use a shallow structural model as $f_\lambda$. Specifically, for HoloGNN (MP), $f_\lambda$ simply performs $\boldsymbol{A}^k(\boldsymbol{V}_1 \oplus \mathbb{1}_v)$, for $k$ set to 2. Note that this variant does not have any extra parameters with respect to a standard GNN. For HoloGNN (GNN), $f_\lambda$ consists of a shallow GNN network, with 2 layers and hidden dimension equal to its input dimension. The GNN layer type depends on the dataset, as we will see next. In both cases, we set $T^\star = 8$ for Planetoid and MovieLens datasets, and $T^\star = 10$ for RelBench.

We further defined $\phi_\lambda$ to be elementwise multiplication and implemented $\rho_\lambda$ to average over all steps $t$. This means we do not concatenate the $\boldsymbol{U}_{\lambda,\{v_1,\ldots,v_r\}}$ for different $\lambda$ values; instead, we treat all $\boldsymbol{U}_{t,\{v_1,\ldots,v_r\}}$ as part of the same partition. This decision aims to reduce the dimension $d_r$ of $R(E(\boldsymbol{A}, \boldsymbol{V}_1))_{\{v_1,\ldots,v_r\}}$ (Equation (11)) to match $d$, aligning it to the dimension of other methods.

For NBFNet, we included initial node features by concatenating them in the initial (boundary) condition. We denote the variant that does not include node features as NBFNet (no feat).

Notably, while a standard GNN returns node embeddings, SEAL and NBFNet do not return node embeddings when pre-trained on tasks of order greater than 1. SEAL produces pre-trained embeddings corresponding to the order of the task, which can scale up to $\mathcal{O}(n^3)$ in our experiments when pre-trained on order-3 tasks. Similarly, NBFNet returns order-2 embeddings when pre-trained on tasks of that order and is unable to extend to higher-order tasks. In contrast, HoloGNN remains linear in the number of nodes and can handle tasks of any order, providing a scalable and flexible solution across task complexities.

We train all models using the Adam optimizer, and report the test for the configuration achieving the best validation metric. Each experiment is repeated for 3 different seeds. Details of hyperparameter grid for each dataset can be found in the following paragraphs.

**MovieLens.** For all models except NBFNet, we use a Bipartite-SAGE GNN Network (Hamilton et al., 2017) with hidden dimension 64 and 3 SAGE layers. For SAGE & LapPE we tuned the number of eigenvectors in $\{8, 16\}$. We tuned the learning rate in $\{0.01, 0.001\}$. For HoloGNN (MP) and HoloGNN (GNN), we use the Bipartite-SAGE GNN Network as $g_{\text{struc}}$ (Equation (1)) and use node degree as the breaking rule to construct breaking selectors (Appendix A.1). We employed GCN layers in the GNN used as $f_\lambda$ for HoloGNN (GNN) for all datasets except **UserMovieUser**, where we instead used SAGE layers. We generate train validation and test splits for each task by randomly splitting the edges with a ratio of 80:10:10. We run our experiments for 1k epochs, and return the test at the epoch achieving the best validation results. For NBFNet and SEAL we run for 50 epochs, due to the significant and intractable increase in running time. We remark that the number of epochs for these models is similar to the number of epochs used in the corresponding original papers on their datasets. Notably, on **UserMovieUser** and **UserMovieMovie**, preprocessing for SEAL to construct the subgraphs exceeds the RAM capacity of our A100 GPUs (528 GB), resulting in out-of-memory (OOM) errors as reported in the table. Finally, since NBFNet is not designed for triplet predictions, we report NA in the corresponding entries for **UserMovieUser** and **UserMovieMovie**.

**Planetoid.** For all models except NBFNet, we use a GNN Network with hidden dimension 64 and 2 GCN layers (Kipf & Welling, 2017). For GCN & LapPE we tuned the number of eigenvectors in $\{8, 16\}$. For the node-level tasks, we employ the widely used choices, including the final classifier being a linear layer, dropout of $0.5$ and weight decay of $5e - 4$. For HoloGNN (MP) and HoloGNN (GNN), we use the GCN network as $g_{\text{struc}}$ (Equation (1)), node degree as the breaking rule to construct breaking selectors (Appendix A.1), and we further employed GCN layers in the GNN used as $f_\lambda$ for HoloGNN (GNN). For node classification, we employ the standard splits (Yang et al., 2016), while for link prediction, following Zhu et al. (2021), we generate train validation and test splits by randomly splitting the edges with a ratio of 85:5:10. We run our experiments for 500 epochs, and return the test at the epoch achieving the best validation results. Due to the significant increase in running times, for NBFNet we run for 100 epochs for the node-level tasks, and for 20 epochs for the link-level tasks, while for SEAL we run for 500 and 50, respectively.

**RelBench.** The `rel-stack` graph is a large and realistic graph containing over 38M nodes. Because of this, our models use subgraph-based sampling as proposed by Hamilton et al. (2017). This means that the symmetry breaking approaches based on global structural properties, such as choosing nodes of highest-degree, are not possible in this case. Instead, we use a mini-batch based breaking method, which takes the embeddings of each node given by the initial structural model $g_{\text{struc}}$ and passes them through a learnable linear layer to produce a single scalar logic for each node. These are then passed as input to a Gumbel-Softmax computation block as described in Appendix A.1, which selects $T^\star = 10 \ll 38M$ distinct nodes to be used as breaking nodes in a fully differentiable way. We adapt the GNN training code from the RelBench repository found at https://github.com/snap-stanford/relbench. We adopt many default hyperparameters from the repository, including 10 epochs training for node-level tasks, and 20 for link prediction, 2 GNN message passing layers for $f_t$ with sum aggregation, and Adam optimizer with learning rate 0.005 for node-level and 0.001 for link-level tasks. The exception is that we shrunk the network so as to fit on a 24G GPU. Namely we used batch size 128, hidden dimension 64, and sampled 64 neighbors per node.

**Eigenvector Task.** We use the GCN model implemented in Lim et al. (2023), denoted by GCN (constant input) in Table 2, as $g_{\text{struc}}$ (Equation (1)). Rather than training $g_{\text{struc}}$ together with our expansion map and the classifier, we utilize the *pre-trained* $g_{\text{struc}}$ to obtain $V_1$. Then, we use $V_1$ to derive the eigenvectors as per Algorithm 1, thereby only training the final classifier. We obtained the eigenvectors using the Subspace Iteration (Saad, 2011), a generalization of the power iteration method for computing several dominant eigenvectors simultaneously, using node degrees to obtain breaking selectors (Appendix A.1). Given the eigenvectors matrix $U \in \mathbb{R}^{n \times k}$, we obtain the final prediction as $\text{MLP}(U_{i,:} \odot U_{j,:})$, with $\odot$ denoting element-wise product. We set $k$ to be the same as the number of eigenvectors used by the other methods, namely 16.

**Molhiv.** We consider a GNN Network with hidden dimension 64 and 2 GCN layers (Kipf & Welling, 2017) as the baseline. For HoloGNN, We use the identity as $g_{\text{struc}}$ (Equation (1)), and choose $T = 2$ breakings for each graph in the dataset. We further employ 2 GCN layers with hidden dimension 64 in the GNN used as $f_\lambda$. For both models, we tuned the learning rate in $\{0.01, 0.001\}$. For graph classification, we employ the standard splits (Hu et al., 2020), while for link prediction, we generate train validation and test splits by randomly splitting the edges with a ratio of 85:5:10.

