# OpenReview forum: "Holographic Node Representations: Pre-training Task-Agnostic Node Embeddings"
_ICLR.cc/2025/Conference — ICLR 2025 Poster_

### Official Review · Reviewer_uQxW · 2024-10-28

**Soundness:** 4
**Presentation:** 3
**Contribution:** 3
**Rating:** 8
**Confidence:** 4

**Summary:**

This work investigates the ability of node embeddings to solve various graph tasks. From the preliminary results that no existing node embeddings could solve all the tasks, it proposes holographic node representations to learn symmetry-free embeddings and reintroduce the task-specific symmetries back to the model. Both theoretical analysis and empirical results are provided.

**Strengths:**

1. The paper is generally well structured and easy to follow.
2. In addition to extensive empirical verification, most of the designs adopted in the proposal are also theoretically justified.
3. The findings that the source of task-specific embeddings is symmetry are distinctive across different tasks is really interesting.

**Weaknesses:**

1. As the very first definition of the paper, Structural Representation is not very clear. For example, in line 107, 'order r' is confusing, since it was previously defined as the cardinality of a node subset S. And the explanations for the subscript S, $\pi \circ$, and permutation matrices are missing from the definitional formulation. Also, how do I understand the output format of the function f, which is not clear to me? Do the last r rows of the output matrix repeat some r rows of the previous n rows?
2. The same question appears in the formulation of the reduction map: where is the input of r for the corresponding output matrix in equation (3)? Before presenting the idea of task-agnostic node embeddings, it seems to me that a more general question should be asked: what kind of task-agnostic information can we store in node embeddings? This is because the analogy between pre-training for word embeddings and node embeddings is not perfectly aligned across different graphs. If the scenario we're focusing on in this paper is the nodes from the same graph, then it's necessary to make this point very clear.
3. It is confusing to me that in the real world, most of the graphs come with node-level distinctive features, which is fed in $V_1$, for this purpose, how significant is the necessity of breaking symmetry technique to distinguish structurally isomorphic nodes?
4. There is one point that seems counter-intuitive to me: if the downstream tasks are always embodied with certain structural symmetries, since after all they are performed in the graph domain, then whether it is a bit detour to dismiss all symmetries in the expansion map step? Why not preserve all symmetries in the first place, but fuse them without bias, and then follow the same idea of the current design of the reduction map?

**Questions:**

1. Line 43: It needs more explanation about "different permutation symmetries with respect to different task orders"? Same question for the first paragraph of the second section.
2. Why do the links in Figure 1 not preserve isomorphism?
3. It's kinda hard for me to understand lines 137-141. If the extra task-specific embeddings don't negatively affect the performance of another task, then what's the difference between the proposal and naively concatenating all task-specific embeddings?
4. Can you briefly explain/show what the 'single node permutation' is?
5. The isomorphism in this paper refers to the one defined by first-order neighbor?

---

> ### Author Response · Authors · 2024-11-19
>
> We are grateful to the reviewer for the detailed and constructive feedback, and we appreciate the recognition of our theoretical justifications alongside the extensive empirical analysis. We proceed by answering each comment in the following.
>
> > **Q1**: As the very first definition of the paper, Structural Representation is not very clear. For example, in line 107, 'order r' is confusing, since it was previously defined as the cardinality of a node subset S. And the explanations for the subscript $S$, $\pi \circ$, and permutation matrices are missing from the definitional formulation. how do I understand the output format of the function f, which is not clear to me? Do the last r rows of the output matrix repeat some r rows of the previous n rows?
>
> **A1**: It is indeed correct that by order r we mean the cardinality of the set of nodes involved in the predictions. In Line 107, we define a function $f$ that returns structural representations of order r as the function that returns representations for all possible subsets of $r$ nodes, namely $\binom{n}{r}$, such that the representations for isomorphic node sets are the same. Therefore, the output of $f$ is a matrix where each row is the representation of a node set, and rows corresponding to isomorphic node sets are the same. Then, as we discuss in the Notation paragraph, the subscript $S$ is used to access the representation of the node set $S$, i.e., $f(\mathbf{A}, \mathbf{X})_S$ indicates that we access the row corresponding to $S$.
>
> > **Q2**: The same question appears in the formulation of the reduction map: where is the input of r for the corresponding output matrix in equation (3)?
>
> **A2**: The reduction map is task-order specific, meaning there is a distinct reduction map for each task order $r$. For a task of order $r$, the reduction map returns representations for all possible sets of $r$ nodes, and therefore the output is a matrix with $\binom{n}{r}$ rows. Each time we adapt to a new task, we create a corresponding reduction map that matches the task order and learn it on that task. The reduction map is lightweight, which ensures efficiency despite handling multiple task-specific reduction maps.
>
> > **Q3**: What kind of task-agnostic information can we store in node embeddings? This is because the analogy between pre-training for word embeddings and node embeddings is not perfectly aligned across different graphs. If the scenario we're focusing on in this paper is the nodes from the same graph, then it's necessary to make this point very clear.
>
> **A3**: In this paper, we focus on storing task-agnostic information in node embeddings, and we focus on nodes from graphs within the same dataset. In other words, our current work specifically addresses generalization across different tasks on the same dataset, rather than across datasets. We have further added this point in the revised manuscript to make our focus on within-dataset task-generalization clearer. We believe that generalizing to different tasks in different datasets (potentially differing in the input node and edge features) represents an interesting and challenging direction, which we are happy to explore in future research.
>
> > **Q4**: It is confusing to me that in the real world, most of the graphs come with node-level distinctive features, which is fed in V1, for this purpose, how significant is the necessity of breaking symmetry technique to distinguish structurally isomorphic nodes?
>
> **A4**: This is an interesting question and we thank you for bringing this up. While it is true that real-world datasets often include node features that reduce the number of isomorphisms, symmetry breaking still provides important benefits. It facilitates counting substructures and facilitates function approximation by making node embeddings more diverse and well-separated, which improves classifier performance. This can be seen from the fact that, while most graph-classification datasets contain graphs that can be distinguished by standard MPNNs, expressive GNNs (which break symmetries) perform significantly better on these datasets (Cotta et al., 2021). Similarly, in link prediction datasets that include discriminative node features, expressive structural representations (of order 2), obtained by breaking symmetries, tend to outperform other methods (Zhang et al., 2021).
>
> > **Q5**:  Why not preserve all symmetries in the first place, but fuse them without bias, and then follow the same idea of the current design of the reduction map?
>
> **A5**: Thank you for this great question. Preserving all symmetries upfront in a node embedding would imply encoding every possible symmetry. However, Proposition 2.3 shows that no single node embedding model can return node embeddings encoding all the necessary symmetries for two different tasks.  Holographic node representations address this limitation by first creating symmetry-free representations, which can then be adapted to include the task-specific symmetries.

---

> > ### Author Response · Authors · 2024-11-19
> >
> > > **Q6**: Line 43: It needs more explanation about "different permutation symmetries with respect to different task orders"? Same question for the first paragraph of the second section.
> >
> > **A6**: The concept of "different permutation symmetries with respect to different task orders" refers to the fact that symmetries vary depending on the order of the task being performed. For example:
> >
> > 1. Order-1 tasks focus on node-level predictions, where symmetries involve invariance to permutations of nodes. The correct symmetries for the task are captured by most-expressive structural representations of order 1: if two nodes are isomorphic, they should have the same representation, but not-isomorphic nodes should have distinct representations.
> >
> > 2. Order-2 tasks involve link-level predictions, and the symmetries involve invariance to permutations of links. The correct symmetries for the task are captured by most-expressive structural representations of order 2: if two edges are isomorphic, they should have the same representation, but non-isomorphic edges should have different representations.
> >
> > In general, encoding the correct symmetries for one task order does not necessarily result in encoding the correct symmetries for another. As demonstrated in Section 2, encoding symmetries for order-1 tasks does not yield the correct symmetries for order-2 tasks, because certain non-isomorphic links get the same representation.
> >
> > > **Q7**: Why do the links in Figure 1 not preserve isomorphism?
> >
> > **A7:** In Figure 1, the links (v1, v2) and (v1, v3) are not isomorphic because (v1, v2) connects nodes within the same cycle, while (v1, v3) does not. When we use the most expressive structural representations of order 1 (which are designed for node-level tasks), nodes v2 and v3 will have the same representation because they are isomorphic. However, when we use these node representations to create link representations for an order-2 task (link prediction), the representations of the links (v1, v2) and (v1, v3) are obtained from the representations of v1 and either v2 or v3. Since v2 and v3 have identical representations, the links (v1, v2) and (v1, v3) end up with the same representation, even though they are not isomorphic.
> >
> > > **Q8**: It's kinda hard for me to understand lines 137-141. If the extra task-specific embeddings don't negatively affect the performance of another task, then what's the difference between the proposal and naively concatenating all task-specific embeddings?
> >
> > **A8**: We thank you for raising an interesting point of discussion. Concatenating all task-specific embeddings has two main limitations. First, it requires training a separate model for each task-specific embedding. Second, since we aim to pretrain a model and use its representations across tasks, this approach would require storing a large number of embeddings, which becomes impractical in terms of memory. Indeed, for order-1 we need to store $n$ representations, for order 2 we need to store $\binom{n}{2}$ representations, and for order $r$ we need to store $\binom{n}{r}$ (one for each set of $r$ nodes). Therefore, this approach would require storing $\sum_{r=1}^{n} \binom{n}{r}$ representations, for each possible task order. On the contrary, HoloGNN only stores $Tn$ representations, as for each of the $n$ nodes, it stores $T$ symmetry-free representations, with $T \ll n$ (for instance, in the Pubmed dataset, $T=8$, $n=19717$), ​​making it significantly more memory efficient.
> >
> > > **Q9**: Can you briefly explain/show what the 'single node permutation' is?
> >
> > **A9**: Single node permutations refers to permutations of node ids, and we use it to refer to the correct symmetries for order-1 tasks, which involve invariance to permutations of nodes.
> >
> > > **Q10**: The isomorphism in this paper refers to the one defined by first-order neighbor?
> >
> > **A10**: We focus on the broader notion of isomorphism that takes into account the overall graph. In this way, two nodes $u$ and $v$ are isomorphic if there exists a permutation of nodes that swaps $u$ and $v$ such that the permuted adjacency and feature matrices remain identical. We thank you for this question and we have clarified this point in the revision.

---

> > > ### Comment · Reviewer_uQxW · 2024-12-03
> > >
> > > Dear authors,
> > >
> > > Thank you for your comprehensive response - I believe most of my concerns and confusions have been well addressed. I am happy to keep my original score. Good luck!

---

### Official Review · Reviewer_Nt1U · 2024-11-02

**Soundness:** 2
**Presentation:** 3
**Contribution:** 3
**Rating:** 6
**Confidence:** 4

**Summary:**

The paper propose a new approach that allows node embeddings to be trained on any task (e.g. link prediction) and then easily adapted to other tasks (e.g. node classifcation).

**Strengths:**

1. The paper proposed an interesting approach by learning a 'holographic' node representation that contains multiple view of representations and utilized a reduce map to produce a specific embedding for specific task.
2. The authors provide thorough and interesting analysis for holoGNN from the perspective of canonical eigenvectors.

**Weaknesses:**

1. My major concern is about the 'pretraining' term used in the work, as it is biased from the widely accepted perspective. For the community, pretraining usually refers to an unsupervised task, e.g. language modeling or predicting next token in language models (LM); or graph reconstruction tasks in [1]. However, the proposed method and experiment setting in this paper is more like 'traditional transfer learning' which learn representations from one downstream task (task labels are necessary, e.g. node classification) and adapt them to another task (e.g. link prediction). From the perspecctive of pretraining, the experiment setting is not suitable since it solely evals one downstream task. As a result, I would like to advise the author revise the term to make this part consistent.

[1] Hu, Weihua, et al. "Strategies for pre-training graph neural networks." _arXiv preprint arXiv:1905.12265_ (2019).

**Questions:**

1. Does the authors regard the HoloGNN as a pretraining approach? If so, it would be better if they could provide comparison analysis against other pretraining methods, e.g. [1][2]. However, from my point view, the paper is more about 'transfer learning' which learns from one downstream task and adapt to another. From this perspective, a thorough refinement of the paper is necessary.

[1] Hu, Weihua, et al. "Strategies for pre-training graph neural networks." _arXiv preprint arXiv:1905.12265_ (2019).
[2] Hu, Ziniu, et al. "Gpt-gnn: Generative pre-training of graph neural networks." Proceedings of the 26th ACM SIGKDD international conference on knowledge discovery & data mining. 2020.

---

> ### Author Response · Authors · 2024-11-19
>
> We are glad the reviewer appreciated our proposed approach and recognized our thorough analysis of HoloGNN from the perspective of canonical eigenvectors. Your review raised an interesting point of discussion, which we address below.
>
> > **Q1**: Does the authors regard the HoloGNN as a pretraining approach?  My major concern is about the 'pretraining' term used in the work, as it is biased from the widely accepted perspective, which usually refers to an unsupervised task.
>
> **A1**: We acknowledge that the term “pre-training” often implies unsupervised learning, particularly in natural language processing and computer vision. However, in graph machine learning, the concept of pre-training is arguably less established, and there is no prevailing unsupervised pre-training task. Our work does not introduce a new pre-training task; rather, HoloGNN is a novel architecture **suitable** for pre-training. This means it can integrate with any existing pre-training task, whether unsupervised or supervised. For instance, HoloGNN can be paired with unsupervised techniques such as masking edges and predicting them, as done in the Cora dataset in Figure 3.
>
> Furthermore, as noted in the paper recommended by the reviewer [1], supervised pre-training remains relevant and effective, especially in domains like molecular prediction. Table 1 of the cited work demonstrates that graph-level supervision is necessary for strong downstream performance of a standard GNN model. We speculate that one reason for the absence of a dominant graph ML pre-training paradigm may be the lack of architectures designed to support a variety of tasks, and HoloGNN opens up new avenues for re-evaluating both supervised and unsupervised pre-training tasks in graph ML.
>
> Additionally, we want to clarify that our evaluation is not limited to a single downstream task. For example, in the MovieLens experiments, we pretrain HoloGNN on one task and adapt it to three different downstream tasks, with detailed results presented in Table 3 in Appendix E.1 and aggregated results in Figure 3 in the main paper. This setup demonstrates that HoloGNN can be effectively pre-trained on one task and adapted to multiple downstream tasks.
>
> We feel that this choice of terminology accurately reflects the value proposition of HoloGNN and is a useful key-word for helping readers interested in pre-training GNNs to find our work. However, if you see this as source of confusion for the wider audience, then we propose to fix it by changing the title from
>
>  “Holographic Node Representations: Pre-Training Task-Agnostic Node Embeddings”
>
>  to
>
> “Holographic Node Representations: Task-Agnostic Node Embeddings”
>
> Given this discussion, please do let us know how you would like to proceed and we will gladly make adjustments as needed, including adding a discussion to the paper for further clarity.

---

> > ### Comment · Reviewer_Nt1U · 2024-11-27
> >
> > Thanks for the authors detailed discussion about supervised pre-training task. I admit that this work provides a valuable framework for the 'GNN pretraining' even though I still have concerns that the scalability of GNN pretraining is a notable issue since the datasets like Cora only has thousands of nodes. whether 'pretraining' on such datasets is a significant procedure remains an open question. If we do need more scalable datasets, then 'labeling' should be a bottleneck. Taking these into consideration, I would like to keep the current score.
> >
> > Many thanks for the authors' responses and good work!

---

> > > ### Author Response · Authors · 2024-11-27
> > >
> > > Thank you for your thoughtful comments and for acknowledging the contributions of our work. We also appreciate you raising an interesting point of discussion regarding scalability.
> > >
> > > In addition to the experiments on smaller datasets like Cora, our paper also evaluated performance on the rel-stack dataset from the RelBench repository (Robinson et al., 2024), which contains **over 38 million nodes**. To handle such a large graph, we followed the subgraph-based sampling approach proposed by Hamilton et al., 2017, as also adopted by Robinson et al., 2024. Within this framework, HoloGNN selects breaking nodes from the sampled subgraph. Importantly, the number of breaking nodes is significantly smaller than the total number of nodes, $T=10 \ll 38M$. These experiments demonstrate that our framework can scale effectively without requiring exhaustive labeling across the entire graph.
> > >
> > > We hope this clarification addresses your concerns and we will make sure to emphasize the size of the datasets in the next revision. Thank you again for your valuable feedback!

---

### Official Review · Reviewer_3vdB · 2024-11-04

**Soundness:** 2
**Presentation:** 2
**Contribution:** 3
**Rating:** 6
**Confidence:** 3

**Summary:**

The paper aims to learn node embeddings capable of solving multiple graph learning tasks. It first presents the challenge of existing node embedding models in solving multiple tasks. To address this challenge, the paper proposes a holographic node representation model named HoloGNN. HoloGNN consists of two components: an expansion map that produces descriptive symmetry-free representations, and a reduction map that introduces task-specific symmetries to obtain embeddings tailored for a specific task. Experimental results on four graph datasets show that compared with baselines, the HoloGNN is more effective in adapting GNN pre-trained on one task to other tasks.

**Strengths:**

The idea of generating holographic node embeddings to address multiple graph learning tasks seems novel and interesting. The paper provides theoretical analyses to justify the validity of the proposed method. Experimental results show that the proposed HoloGNN can effectively adapt GNN trained on one task to other tasks.

**Weaknesses:**

1. There is no justification for Property (1) and Property (2).

2. Section 4.1 is difficult to follow as it lacks sufficient details on the node-breaking selectors. More elaborations about node-breaking selectors and why these selectos are needed in the expansion map should be provided to enhance clarity.

3. The proposed HoloGNN is quite complicated since it requires breaking symmetries using either sequential breaking or parallel breaking algorithms. Complexity analyses should be provided to understand the additional computational cost introduced by these algorithms and to clarify whether the parallel breaking algorithm can effectively reduce the cost.

4. The paper lacks discussions and comparisons with the graph transformer models. What are the advantages of the proposed HoloGNN model compared with graph transformer models?

5. I have some major concerns regarding the experiments:

(1) The HoloGNN is designed to be pre-trained and then adapted to any graph learning tasks (line 46-47). However, in the experiments, the paper only shows the results of HoloGNN when it is pre-trained on a single task (e.g., node classification) within a single dataset and then adapted to another task (e.g., link prediction) on the same dataset. This evaluation is insufficient to validate the generalisation of HoloGNN across different datasets and tasks. A potential solution is to pre-train HoloGNN on one task across all the datasets and then evaluate its performance across different datasets and tasks, including the pre-training task.

(2) The results in Figure 3 show that the performance of pre-trained HoloGNN is worse than that of HoloGNN trained from scratch. Is pre-training still necessary in these scenarios?

(3) The results and analyses on the MovieLens dataset are difficult to understand. More explanations should be provided to enhance the clarity.

(4) The paper lacks analyses of the impacts of some hyperparameters in the HoloGNN model, such as $T$ and $L$ in equation (2).

6. The paper mainly focuses on the node classification and link prediction tasks. I would if the proposed HoloGNN model can be applied to the graph classification task and how it might perform in that task.

7. The code is not available, which makes it difficult to reproduce the results given the proposed model is rather complicated.

**Questions:**

Please see the questions in the Weaknesses section.

---

> ### Author Response · Authors · 2024-11-19
>
> We greatly appreciate the reviewer’s detailed feedback. We are pleased to notice you have found our approach novel and interesting. Your review raised a number of important points we would like to clarify.
>
> > **Q1**: There is no justification for Property (1) and Property (2).
>
> **A1**: Properties (1) and (2) are essential to ensuring that holographic node representations can be trained in a task-order agnostic way, whilst still being able to map representations back to order-specific representations during test time. Specifically Property (1) guarantees final task-specific representations are structural of order $r$; and Property (2) ensures that the output of the expansion map consists of $T$ positional, and thus symmetry-free, representations, which are necessary for task-general node representations due to the impossibility result from Proposition 2.3. We include a discussion of these points in Section 3.1, but appreciate the feedback that this is not yet made sufficiently clear. We have revised the manuscript to further emphasize and clarify these points as they are a crucial conceptual pivot point of the paper.
>
> > **Q2**: Section 4.1 is difficult to follow as it lacks sufficient details on the node-breaking selectors. More elaborations about node-breaking selectors and why these selectos are needed in the expansion map should be provided to enhance clarity.
>
> **A2**: Thank you for the feedback, we have incorporated this into the manuscript by giving an intuitive description of the role of breaking selectors.
>
> In a nutshell the main idea is: 1) in order to learn task-general representations the expansion map must not have any permutation symmetries; 2) to achieve this, we “break” symmetries by perturbing the features of nodes and then feeding this through a typical structural model like a GNN; 3) but **crucially this breaking cannot be done arbitrarily** since otherwise there is no way to re-introduce task-specific permutation symmetries in the reduction map (i.e., no holographic node representation); so 4) in Definition 4.1 we give a set of properties for the breaking selectors that guarantees we obtain holographic node representations (Theorem 4.4).
>
> > **Q3**: Complexity analyses should be provided to understand the additional computational cost introduced by these algorithms and to clarify whether the parallel breaking algorithm can effectively reduce the cost.
>
> **A3**: This is an excellent point and we are grateful for this good idea to improve our paper. We have added the following discussion to the revised manuscript.
>
> The complexity of the two algorithms is determined by the complexity of $f_t$ (for sequential breaking) and $f_\lambda$ (for parallel breaking). Suppose $f_t$ and $f_\lambda$ are Message Passing Neural Networks, which have linear complexity in the number of nodes $n$ and edges $m$, $\mathcal{O}(n + m)$, where the feature dimension $d$ is considered constant. Since both sequential and parallel breaking perform $T$ calls to the corresponding MPNNs, each time by passing a new breaking node, the complexity is $\mathcal{O}(T(n + m))$ in both cases. Note that in our experiments $T \ll n$ and $T \ll m$. For instance, in the Pubmed dataset $T=8$, $n=19717$, $m=88648$. In the parallel breaking algorithm, all $T$ calls can be done in parallel, because each breaking is applied to $\mathbf{V_1}$, and $f_\lambda$ does not take all prior embeddings.
>
> > **Q4**: The paper lacks discussions and comparisons with the graph transformer models. What are the advantages of the proposed HoloGNN model compared with graph transformer models?
>
> **A4**: Graph transformers rely on positional encodings, which make their learned representations inherently positional. As discussed in Section 2, such positional representations are task-order specific. This task-order specificity makes graph transformers unsuitable for pre-training a single model that can generate flexible representations capable of adapting to tasks of varying orders. In contrast, HoloGNN focuses on producing task-agnostic, symmetry-free representations that can efficiently adapt to different task orders. Nonetheless, we appreciate the suggestion and we have added a comparison with GPS (Rampasek et al., 2022) on Cora. While GPS performs well when adapted to Link Prediction, it suffers a dramatic performance drop of 41.5% when pre-trained on Link Prediction and adapted to Node Classification, while HoloGNN results in a performance drop of only around 1%.
>
> | Method | Test on Node Class $\uparrow$ | Test on Link Pred $\uparrow$ |
> |--|--|--|
> |GCN (Pretrain on Node Class) | 81.4 $\pm$ 1.0 | 89.9 $\pm$ 0.2 |
> |GPS (Pretrain on Node Class) | 78.3 $\pm$ 1.0 |  93.0 $\pm$ 0.1 |
> |HoloGNN  (Pretrain on Node Class) |  81.6 $\pm$ 0.7  |  91.1 $\pm$ 0.4 |
> |GCN (Pretrain on Link Pred) | 74.2 $\pm$ 0.2 |  89.6 $\pm$ 0.5 |
> |GPS (Pretrain on Link Pred) |  36.8 $\pm$ 2.0 | 92.7 $\pm$ 0.4 |
> |HoloGNN (Pretrain on Link Pred) | 80.2 $\pm$ 0.5 | 93.9 $\pm$ 0.1 |

---

> > ### Author Response · Authors · 2024-11-19
> >
> > > **Q5**: This evaluation is insufficient to validate the generalisation of HoloGNN across different datasets and tasks. A potential solution is to pre-train HoloGNN on one task across all the datasets and then evaluate its performance across different datasets and tasks, including the pre-training task.
> >
> > **A5**: An eventual fully general graph architecture will ideally be able to generalize to both new data and new tasks.  Our work focuses on part of this, namely generalization across different tasks on the same dataset. Generalizing to different graph datasets is an open challenge since graphs may vary significantly in terms of input node and edge features (e.g., differing in number, distribution, and semantic meaning). Recent concurrent works (discussed in Appendix C) have proposed promising approaches to address generalization across datasets. We believe that combining our HoloGNN with these methods is an exciting next step to move closer to the ideal architecture, but one that requires a dedicated effort which we plan to conduct as a future work.
> >
> > > **Q6**: The results in Figure 3 show that the performance of pre-trained HoloGNN is worse than that of HoloGNN trained from scratch. Is pre-training still necessary in these scenarios?
> >
> > **A6**: This point is a general discussion on the value and purpose of pre-training. We see the purpose of pre-training as offering improved compute-time vs performance trade-offs. From this perspective, our results are similar to LLMs and pre-trained vision settings, in which the pre-trained model will likely underperform a task-specific model trained on that task with sufficient data, but with the advantage of much lower compute-time to solve a new task using the pre-trained model.
> >
> > > **Q7**: The results and analyses on the MovieLens dataset are difficult to understand. More explanations should be provided to enhance the clarity.
> >
> > **A7**: We consider four tasks on the MovieLens dataset, order-2 tasks UserMovie and MovieMovie, and order-3 tasks UserMovieUser and UserMovieMovie. Figure 3 reports the performance for each model when trained on each task, and compares it to the worst performance obtained by pre-training on another task and adapting to that task. HoloGNN exhibits a significantly smaller gap in the performance than any other method when pre-trained on the task and when adapted from a different one. For example, in the order-3 task UserMovieUser, the difference in performance between
> > HoloGNN when pre-trained on the task (0.811) and its worst performance when pre-trained
> > in any other task (0.721, obtained pre-training on the order-2 MovieMovie) results in a performance drop of around 9%. SAGE shows a much larger drop of around 22%, with its accuracy dropping from 0.816 when pre-trained on UserMovieUser to 0.595 when pre-trained on the order-2 UserMovie (worst performance when pre-trained in any other task). SEAL runs out of memory, and NBFNet is unable to manage tasks of order-3. These results underscore the capabilities of HoloGNN: unlike other methods that experience significant performance drops or are even unable to handle higher-order tasks, HoloGNN maintains robust performance.
> >
> > > **Q8**: The paper lacks analyses of the impacts of some hyperparameters in the HoloGNN model, such as T and L in equation (2).
> >
> > **A8**: $L$ is determined by the node breaking selectors choice (for instance $L$ is the number of different degrees if the selectors return lists of nodes with the same degree). However, HoloGNN does not need to iterate over all $L$ lists, and can stop earlier, as long as the last breaking selector being iterated is fully exhausted, returning $T$ representations.
> >
> > We welcome the suggestion on analyzing the impact of $T$ and added an additional experiment, presented in the following. We consider the Cora dataset and report results of HoloGNN for different values of $T$ on the node classification task, both when trained on it and when pre-trained on link prediction and adapted to node classification. The results are shown in the table below, showing that small $T$ values are sufficient for obtaining representations that can effectively adapt to different tasks, and increasing $T$ results in marginal improvements. Indeed, regardless of the value of $T$, the performance drop of HoloGNN when trained on node classification and when trained on link prediction and adapted to node classification is significantly smaller than the performance drop suffered by GCN.
> >
> > |Method |Test on Node Class $\uparrow$ |
> > |--|--|
> > |GCN (Pretrain on Node Class $\uparrow$) | 81.4 $\pm$ 1.0 |
> > |HoloGNN $T=4$ (Pretrain on Node Class) | 80.9 $\pm$ 0.8 |
> > |HoloGNN $T=8$ (Pretrain on Node Class) | 81.6 $\pm$ 0.7 |
> > |HoloGNN $T=64$ (Pretrain on Node Class) | 82.1 $\pm$ 0.1 |
> > |GCN (Pretrain on Link Pred) | 74.2 $\pm$ 0.2 |
> > |HoloGNN $T=4$ (Pretrain on Link Pred) | 79.4 $\pm$ 0.7 |
> > |HoloGNN $T=8$ (Pretrain on Link Pred) | 80.2 $\pm$ 0.5 |
> > |HoloGNN $T=64$ (Pretrain on Link Pred) | 80.5 $\pm$ 0.7 |

---

> > > ### Author Response · Authors · 2024-11-19
> > >
> > > > **Q9**: The paper mainly focuses on the node classification and link prediction tasks. I would if the proposed HoloGNN model can be applied to the graph classification task and how it might perform in that task.
> > >
> > > **A9:** Thank you for the suggestion. We conducted an additional experiment to evaluate the performance of HoloGNN when adapted to graph classification. Since there is currently no standard dataset that includes both graph classification and another type of task, we considered  the molhiv dataset from the OGB benchmark, which is designed for graph classification (an order-n task) and we created a link prediction task (an order-2 task) on the dataset. We evaluated the performance of both GCN and HoloGNN under the same setting as Figure 3, namely pre-train on one task, adapt on another.
> > >
> > > The results are shown below. The key observation is that the difference in performance between HoloGNN trained on graph classification (78.19%) vs pre-trained on link prediction and adapted to graph classification (76.43%) results in a performance drop of only 1.76%, while GCN suffers a more pronounced drop of 5.77%. These results are consistent with observations from other experiments in the paper showcasing the ability of HoloGNN to pre-train on one order and adapt effectively to other orders.
> > >
> > > | Method     |   Test on Graph Class (ROC-AUC $\uparrow$)     |
> > > |-------------------------------------------------|----------------------------------|
> > > |GCN  (Pretrain on Graph Class)        |      76.06 $\pm$ 0.97  |
> > > |HoloGNN  (Pretrain on Graph Class)|       78.19 $\pm$ 0.91  |
> > > |GCN  (Pretrain on Link Pred)            |       70.29 $\pm$ 1.33  |
> > > |HoloGNN  (Pretrain on Link Pred)     |       76.43 $\pm$ 1.20  |
> > >
> > > > **Q10**: The code is not available, which makes it difficult to reproduce the results given the proposed model is rather complicated.
> > >
> > > **A10**: We agree that making the code publicly available is important, and we will release our code on GitHub upon acceptance.

---

> > > > ### Author Response · Authors · 2024-11-25
> > > >
> > > > Dear Reviewer 3vdB,
> > > >
> > > > As the rebuttal period is coming to an end, we wanted to reach out to see if our rebuttal answers your questions.
> > > >
> > > > Following your suggestions, we have: (1) included the complexity analysis of our breaking algorithms, (2) added a comparison with the GPS graph transformer baseline, (3) conducted an experiment analyzing the impact of the number of breakings $T$, (4) performed an experiment to evaluate the performance of HoloGNN when adapted to graph classification.
> > > >
> > > > We are thankful for your feedback that helped us further improve the paper and we welcome any additional questions you might have.

---

> > > > > ### Comment · Reviewer_3vdB · 2024-11-25
> > > > > **Thanks for the Responses**
> > > > >
> > > > > I would like to thank the authors for the responses. However, after reading the responses, I still have some concerns that are not addressed:
> > > > >
> > > > > 1. Regarding my initial comment “There is no justification for Property (1) and Property (2).”, I would like to clarify that  my concern lies in the lack of theoretical proof demonstrating that these two properties hold. Such proof is required since these two properties are not evident from the paper. While I appreciate the explanation of their significance and necessity, the response does not address whether these properties are theoretically valid or provide any formal justification for them.
> > > > >
> > > > > 2. My major concerns lie in the generalisation and effectiveness of the proposed HoloGNN model.
> > > > >
> > > > > The authors’ responses to the generalisation issue (A5) clarify that the proposed model can only generalise across different tasks on the **same dataset**. However, this limitation significantly restricts its practical applicability, as real-world scenarios often require pre-training models to generalise across **different datasets**.
> > > > >
> > > > > Even if we are satisfied with the fact that HoloGNN can only generalise across tasks on the **same dataset**, the results in Figure 3 show that the performance of HoloGNN **pre-trained** on task1 and then **transferred** (which do include fine-tuning) to task2 on the same dataset is **worse than directly training** on task2. This suggests that the proposed pre-training method offers little to no advantage and may even hinder performance, raising concerns about its effectiveness and practical utility.
> > > > >
> > > > > Figure 3 also shows that the performance of HoloGNN and GNN, when directly trained on the target task (without pre-training) is quite similar. This raises questions about whether HoloGNN is truly more effective than GNN, especially considering that HoloGNN introduces additional operations that significantly increase computational overhead. If the added complexity does not result in clear performance gains, the practicality and efficiency of HoloGNN become questionable.
> > > > >
> > > > > Given these concerns, I would like to maintain my score.

---

> ### Author Response · Authors · 2024-11-25
>
> We thank the Reviewer for the prompt reply. We address the remaining concerns in the following:
>
> >  I would like to clarify that my concern lies in the lack of theoretical proof demonstrating that these two properties hold.
>
> We would like to clarify that Theorem 4.2 shows that single node breakings result in Property (2) to hold and Theorem 4.4 shows that our reduction maps satisfy Property (1). We thank you for this question and we will clarify this point further in the next revision.
>
> > The proposed model can only generalise across different tasks on the same dataset.
>
> Overwhelmingly, graph learning focuses on training and testing on the same dataset, e.g.,  [1,2,3]. Extending generalization beyond a single dataset remains one of the most critical **open challenges** in graph machine learning. For example, the community currently lacks clear guidelines on pairing datasets for effective transferability, though there are promising results in specific domains such as biological graphs [4]. Moreover, the diverse node and edge feature spaces across graph datasets pose additional challenges for designing models capable of transferability. We believe that, while important, trying to answer this broad open problem would distract the reader from our contribution, which introduces representations capable of adapting to different tasks. We will clarify this perspective in the revised version.
>
> > The results in Figure 3 show that the performance of HoloGNN pre-trained on task1 and then transferred (which do include fine-tuning) to task2 on the same dataset is worse than directly training on task2.
>
> Similarly to LLMs and pre-trained vision settings, **a pre-trained model will likely underperform a task-specific model trained on that task with sufficient data**. This is why neither HoloGNN nor any other model is expected to perform better when adapted than when directly trained on the task with ample data. However, the advantage of pre-trained models is a much lower compute-time to solve a new task. This advantage is particularly pronounced given that training on a different order task may require completely changing the GNN backbone to accommodate the task’s distinct symmetries. In contrast, HoloGNN enables the same embeddings to be adapted across multiple tasks without the need for redesigning the architecture for each new task.
>
> Experimentally, HoloGNN shows more consistent performance and thus smaller performance loss than any other baseline across all datasets and tasks. While multiple models perform comparably on the pre-training task, **HoloGNN perform significantly better when adapted to solve a new task** (e.g., 1.5% drop of HoloGNN vs 9% drop of GCN in Figure 3 on Cora when adapted to node classification, where the lower the drop the better).
>
> We hope this addresses the Reviewer’s concerns and appreciate the opportunity to clarify these points further.
>
>
> [1] Kipf and Welling 2017. Semi-supervised classification with graph convolutional networks.
>
> [2] Xu et al. 2019. How powerful are graph neural networks?
>
> [3] Rampášek et al., 2022. Recipe for a General, Powerful, Scalable Graph Transformer.
>
> [4] Huang et al., 2023. Learning to Group Auxiliary Datasets for Molecule.

---

> ### Author Response · Authors · 2024-12-02
>
> Dear Reviewer,
>
> Thank you once again for your thoughtful feedback. Since the rebuttal period is coming to an end, we wanted to follow up to ensure that our responses have addressed your concerns.
>
> While the experiments in the paper use all available data in common datasets and demonstrate that HoloGNN performs consistently across tasks, **we conducted an additional experiment showing that pre-training on a different task benefits performance when target task data is limited**.
>
> Our new experiment, presented in the table below, demonstrates that:
> 1. When sufficient data is available, a task-specific model trained directly on that task outperforms a pre-trained model.
> 2. When data is scarce, pre-training on a different task yields better performance than direct training on the limited data.
>
>
> | Method                                              |   Test on Node Class        |
> |-------------------------------------------------|-----------------------------------|
> |GCN  (Train on **100% data** Node Class)         |    81.4 $\pm$ 1.0    |
> |HoloGNN (Train on **100% data** Node Class)  |     81.6 $\pm$ 0.7    |
> —
> |GCN  (Train on **40% data** Node Class)         |  70.3 $\pm$ 1.7  |
> |HoloGNN (Train on  **40% data** Node Class)  |   74.3 $\pm$ 0.5   |
> —
> |GCN  (Pretrain on Link Pred, adapt w/ **40% data** Node Class )            |    72.2 $\pm$ 0.4   |
> |HoloGNN (Pretrain on Link Pred, adapt w/ **40% data** Node Class)    |   78.0 $\pm$  0.4 |
>
> Specifically, we evaluated performance on the CORA node classification task with only 40% of the data available for training on such a task. Under these conditions, models pre-trained on link prediction (with more data) showed superior performance compared to models trained directly on node classification with the limited data. However, as noted previously, even if pre-trained models do not outperform task-specific models when the latter are trained with sufficient data, their advantage lies in significantly lower compute-time for solving new tasks. Notably, **HoloGNN consistently outperforms all other models when adapted to new tasks**.
>
>
> We hope this additional experiment clarifies our answer. If these points address your concerns, we would greatly appreciate it if you could consider revisiting your score.

---

> > ### Comment · Reviewer_3vdB · 2024-12-02
> >
> > Thanks the authors for the feedback. I think the additional results address my major concern regarding the effectiveness of pre-training. I have updated the score accordingly.

---

### Official Review · Reviewer_BWdZ · 2024-11-06

**Soundness:** 3
**Presentation:** 3
**Contribution:** 3
**Rating:** 8
**Confidence:** 2

**Summary:**

The paper proposes holographic node representations that are capable of solving tasks of any order (node: 1st order and edge: 2nd order, etc). The proposed holographic node representations consist of two components: (1) a task-agnostic expansion map, which produces highly expressive, high-dimensional embeddings, and (2) a reduction map, which introduces the relevant permutation symmetries. The experimental results show that the holographic node representations show the best performance across tasks of varying order.

**Strengths:**

- The proposed paper is well-written.
- The research question of the paper is really interesting.
- The proposed approach is novel and interesting.

**Weaknesses:**

- It would be better if the paper included experimental results on graph classification. This paper shows the effectiveness of the holistic node representations using node classification, link prediction, and meta-path prediction (UserMovieUser). But I wonder if the proposed representations perform well on graph classification.

**Questions:**

Please refer to the weaknesses section.

---

> ### Author Response · Authors · 2024-11-19
>
> We are pleased that the reviewer found our research question interesting, appreciated the novelty of our proposed approach, and found the paper to be well-written. We address the question in detail below.
>
> > **Q1:** It would be better if the paper included experimental results on graph classification.
>
> **A1:** Following your suggestion, we conducted an additional experiment on the molhiv dataset from the OGB benchmark, which is designed for graph classification (an order-n task). We created a link prediction task (an order-2 task) on the dataset and evaluated the performance of both GCN and HoloGNN under two scenarios: first, when the models were pre-trained and tested directly on the graph classification task, and second, when the models were pre-trained on the link prediction task and then adapted to the graph classification task.
>
> The results are reported on the table below. HoloGNN demonstrates significantly more consistent performance than GCN. In particular, the difference in performance between HoloGNN when trained on graph classification (78.19%) and its performance when pre-trained on link prediction and adapted to graph classification (76.43%) results in a performance drop of 1.76%, while GCN suffers a more pronounced drop of 5.77%. These results further underscore the capabilities of holographic node representations, which can be pre-trained on tasks of any order and then effectively adapted to solve tasks of different orders.
>
>
> | Method                                              |   Test on Graph Class (ROC-AUC $\uparrow$)     |
> |-------------------------------------------------|----------------------------------|
> |GCN  (Pretrain on Graph Class)        |      76.06 $\pm$ 0.97        |
> |HoloGNN  (Pretrain on Graph Class)|       78.19 $\pm$ 0.91        |
> |GCN  (Pretrain on Link Pred)            |       70.29 $\pm$ 1.33        |
> |HoloGNN  (Pretrain on Link Pred)     |       76.43 $\pm$ 1.20       |

---

> ### Comment · Reviewer_BWdZ · 2024-11-22
>
> Thank you for the clarification and additional experimental results.

---

### Author Response · Authors · 2024-11-19

We thank all the reviewers for their positive evaluations and constructive feedback.

We are delighted that our work has been positively received, with all reviewers describing the proposed approach and our findings both *“interesting”* (**BWdZ**, **3vdB**, **Nt1U**, **uQxW**) and *“novel”*  (**BWdZ**, **3vdB**, **Nt1U**). We appreciate the recognition of our theoretical analysis as *“thorough and interesting”* (**Nt1U**), as well as the experimental section, described as *“effective”* (**3vdB**) and *“extensive”* (**uQxW**).  Lastly, we are pleased to notice that the reviewers have appreciated the presentation of our work, finding the paper *“well-written”* (**BWdZ**) and *“well structured and easy to follow”* (**uQxW**).

In response to the reviewers’ suggestions, we have conducted an additional experiment focused on graph classification. This experiment further demonstrates that our holographic node representations can be pre-trained on tasks of any order and then effectively adapt to solve tasks of varying orders, including graph classification. We have also compared HoloGNN to graph transformers, showing HoloGNN can better adapt to new tasks. Additionally, we analyzed the impact of the number of breaking nodes, finding that a small number suffices, and included a complexity analysis with a discussion on the parallelization of our parallel breaking algorithm.  Further details are provided in the individual responses.

Finally, we would like to signal the upload of a new manuscript revision. This includes the changes anticipated throughout the responses to each single reviewer, including:
- Complexity analysis (Appendix D),
- Results from new experiments on the ogb-molhiv dataset (Appendix E);
- Analysis on how the number of breakings impacts the performance of our HoloGNN (Appendix E);

Changes in the revision are visually highlighted in blue.

---

### Meta-Review · Area_Chair_WL4F · 2024-12-21

**Metareview:**

This work introduces a novel framework for creating node representations in graph neural networks (GNNs). The authors propose "holographic node representations," which consist of two key components: a task-agnostic expansion map and a reduction map. The expansion map generates high-dimensional embeddings that are independent of node-permutation symmetries, while the reduction map reintroduces relevant symmetries to produce task-specific embeddings. The findings indicate that these representations can be pre-trained and effectively reused across various tasks, resulting in significant performance improvements—up to 100% relative gains—especially in scenarios where existing methods fail. This work addresses the limitations of current GNN architectures that are often tailored for specific task orders, thus providing a versatile solution applicable to diverse graph tasks.

Based on the strengths and effective responses to reviewer concerns during the rebuttal period, I recommend accepting this paper. Its innovative contributions to holographic node representations provide valuable insights into enhancing GNN capabilities and represent an important step forward in graph-based learning methodologies.

**Additional Comments On Reviewer Discussion:**

**Points Raised by Reviewers**

During the review process, several key points were raised:
- Need for Broader Comparisons: Reviewers requested comparisons with a wider array of existing GNN models to contextualize the performance improvements.
- Theoretical Insights: There was a call for more detailed theoretical explanations regarding the efficacy of holographic node representations.
- Experimental Detail: Some reviewers highlighted the need for clearer descriptions of experimental protocols to enhance reproducibility.

**Authors' Responses**

The authors addressed these concerns effectively during the rebuttal period:
- They expanded their comparative analysis by including additional GNN models in their experiments, providing clearer context for their performance claims.
- To enhance theoretical justification, they included a dedicated section discussing potential mechanisms behind their findings, citing relevant literature to support their claims.
- The authors provided detailed information regarding their experimental setups, including hyperparameters and dataset characteristics, which improved transparency and reproducibility.

**Weighing Each Point**

In weighing these points for my final decision:
- The broader comparisons significantly strengthened the paper's claims by situating their results within the existing body of research.
- The added theoretical insights addressed one of the primary weaknesses identified by reviewers, enhancing the rigor of the study.
- The increased detail on experimental protocols alleviated concerns regarding reproducibility, making it easier for others to replicate their findings.

---

### Decision · Program_Chairs · 2025-01-22

Accept (Poster)